# Aligning What Matters: Masked Latent Adaptation for Text-to-Audio-Video Generation

**Jiyang Zheng[1,2]\***, **Siqi Pan[3]**, **Yu Yao[1]**, **Zhaoqing Wang[1]**,
**Dadong Wang[2]**, **Tongliang Liu[1]†**

[1]Sydney AI Center, The University of Sydney
[2]CSIRO, Data61 [3]Dolby Laboratories
{jzhe5740, zwan6779}@uni.sydney.edu.au
{siqi.pan}@dolby.com   {dadong.wang}@data61.csiro.au
{yu.yao, tongliang.liu}@sydney.edu.au

## Abstract

Text-to-Audio-Video (T2AV) generation aims to produce temporally and semantically aligned visual and auditory content from natural language descriptions. While recent progress in text-to-audio and text-to-video models has improved generation quality within each modality, jointly modeling them remains challenging due to incomplete and asymmetric correspondence: audio often reflects only a subset of the visual scene, and vice versa. Naively enforcing full alignment introduces semantic noise and temporal mismatches. To address this, we propose a novel framework that performs selective cross-modal alignment through a learnable masking mechanism, enabling the model to isolate and align only the shared latent components relevant to both modalities. This mechanism is integrated into an adaptation module that interfaces with pretrained encoders and decoders from latent video and audio diffusion models, preserving their generative capacity with reduced training overhead. Theoretically, we show that our masked objective provably recovers the minimal set of shared latent variables across modalities. Empirically, our method achieves state-of-the-art performance on standard T2AV benchmarks, demonstrating significant improvements in audiovisual synchronization and semantic consistency.

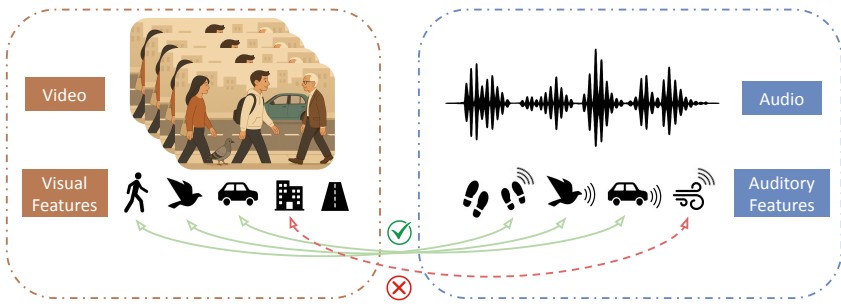

Figure 1: Visual-auditory feature alignment is essential in text-to-audio-video (T2AV) generation, yet assuming full correspondence between audio and visual modalities is often problematic. For example, visual elements like roads or buildings may not produce sound, while audio events such as wind may lack visual presence. Aligning such mismatched features introduces semantic noise, resulting in reduced cross-modal consistency and temporal mismatch in the generated outputs.

---

*Work done during an internship at Dolby.

†Corresponding Author

39th Conference on Neural Information Processing Systems (NeurIPS 2025).

# 1 Introduction

Recent advances in multimodal generative models [2, 24, 28, 37, 39, 48, 42] have enabled high-quality content creation across text, image, audio, and video modalities. While notable progress has been made in text-to-video [3, 10, 21, 13, 38] and text-to-audio [6, 26, 27, 32] generation individually, they are typically studied in isolation, leaving joint audiovisual generation from text largely underexplored. Text-to-Audio-Video (T2AV) generation addresses this gap by aiming to synthesize audio and video streams that are both semantically and temporally aligned, conditioned on a single text prompt. This involves not only generating high-quality content for each modality, but also ensuring that the output audio and video remain contextually consistent and synchronized.

Achieving this requires modeling cross-modal alignment, where both audio and visual representations capture the informative content conveyed by the other modality. To facilitate such alignment, existing approaches often project multimodal features into a shared embedding space [33, 41, 52]. This facilitates the model to capture joint semantics across modalities. However, forcing all audio and visual features to align can be problematic. In real-world settings, audio and visual streams may exhibit only partial alignment: audio may describe only parts of a visual scene, or visual frames may contain elements absent from the audio (See Figure 1). Enforcing full alignment under such conditions introduces mismatched information into the joint representations, resulting in semantically inconsistent or temporally desynchronized outputs during T2AV generation.

To address the challenge of partial correspondence between modalities, we introduce *SAVA*, a framework for **S**elective **A**udio-**V**isual **A**lignment in text-to-audio-video generation. Comparing with existing approaches [31, 33, 41, 45, 52] that assume full alignment between audio and visual features, *SAVA identifies and aligns only those latent components that are jointly predictive across modalities*, while disregarding modality-specific information that could otherwise introduce noise or conflict. The overall pipeline, as shown in Figure 3, proceeds in three stages: *Align and Fine-tune*, it learns to map multimodal latents by selectively filtering out irrelevant dimensions in the latent space using a learnable mask, allowing the model to focus only on features that contribute meaningfully to both modalities. The alignment is learned through adapter networks applied to pretrained encoders. Then, we fine-tune the generator using the aligned multi-condition inputs. *Inference*, it operates in a cascaded manner, projecting features from video and audio into an aligned subspace, and conditioning the corresponding generator on both the text and the aligned video/audio signals. This design enables synchronized and consistent audio-visual generation while preserving efficiency and modularity.

*SAVA* is grounded in a causal view of multimodal generation, where audio and visual signals are generated from a mixture of shared and distinct latent factors. We provably show that the masked alignment objective recovers the minimal set of shared latent variables (those which constitute the true semantic interface between modalities). This not only ensures interpretability and robustness but also mitigates the entanglement issues observed in prior alignment-based models. Our empirical results across diverse benchmarks confirm that *SAVA* significantly improves semantic alignment and temporal synchronization in T2AV generation, outperforming existing baselines. A brief review of related works is provided in Appendix A.

# 2 Problem Formulation

Text-to-Audio-Video (T2AV) generation requires accurate alignment between audio-visual representations to preserve meaningful cross-modal correspondence. In particular, semantic misalignment, where visual and auditory components do not reflect the same underlying content, can mislead generative models and degrade the consistency of the resulting outputs. Our objective is to enable selective and reliable alignment by identifying and preserving only the semantically relevant components across modalities during training. To this end, we begin by reviewing the text-to-audio-video generative process.

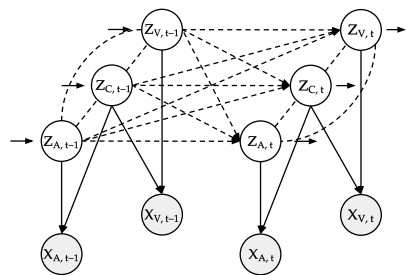

Figure 2: The data generative process of audiovisual data. Audio features $Z_A$ are selectively derived from visual features $Z_v$ guided by a learned mask $m$. Each modality-specific latent combines with residual noise $\epsilon$ to produce the outputs.

**Data Generative Process** As illustrated in Figure 2, we model the audiovisual data generation process using a

structured causal model composed of three latent variable sets: video-specific latent variables $Z_V$, audio-specific latent variables $Z_A$, and cross-modal latent variables $Z_C$, which encode shared content factors underlying both modalities. $Z_C$ may include semantically grounded, temporally evolving entities that manifest in both the visual and auditory domains (e.g., a barking dog or a moving vehicle). In contrast, the modality-specific latents $Z_V$ and $Z_A$ capture factors that are unique to the video and audio domains, respectively. The latent variables are causally connected and evolve over time, with each group at time step $t$ potentially influenced by their own past states and the past states of other groups. Formally, the evolution of these latent variables follows:

$$Z_V^t \leftarrow \{Z_V^{t\text{-}1}, Z_A^{t\text{-}1}, Z_C^{t\text{-}1}\}, \; Z_A^t \leftarrow \{Z_V^{t\text{-}1}, Z_A^{t\text{-}1}, Z_C^{t\text{-}1}\}, \; Z_C^t \leftarrow \{Z_V^{t\text{-}1}, Z_A^{t\text{-}1}, Z_C^{t\text{-}1}\}, \tag{1}$$

where the superscript "past" denotes historical latent states (e.g., from time $t-1$), and the arrows represent causal influence. These relationships reflect the potential bidirectional statistical and causal dependencies [44] across modalities.

The observable variables: video $X_V$ and audio $X_A$, are generated from their corresponding modality-specific latent variables in conjunction with the cross-modal latent:

$$X_V \leftarrow \{Z_V, Z_C\}, \; X_A \leftarrow \{Z_A, Z_C\}. \tag{2}$$

This formulation reflects that while $Z_C$ captures semantically aligned and temporally correlated content, $Z_V$ and $Z_A$ may contain orthogonal information that should not be forced into alignment. Therefore, when attempting to recover cross-modal structure, it is critical to distinguish shared factors from modality-specific ones.

## 3 Methods

**Selective Latent Alignment**   Let $x_v \in X_V$, $X_V \subseteq \mathbb{R}^{T \times H \times W \times 3}$ be a video clip and $x_a \in X_A$, $X_A \subseteq \mathbb{R}^{T' \times M}$ its corresponding audio spectrogram. We extract frozen modality-specific embeddings $\hat{z}_v = f_v(x_v), \hat{z}_a = f_a(x_a)$,, where $f_v$ is a pretrained video VAE encoder [10] and $f_a$ is a pretrained audio diffusion encoder [49, 26]. These raw embeddings may contain modality-specific noise and are not guaranteed to lie in a common semantic subspace. To expose the shared latent structure $Z_C \subseteq \mathbb{R}^d$, we apply learnable reparameterizations (i.e., adapter networks):

$$\tilde{z}_v = q_V(\hat{z}_v), \qquad \tilde{z}_a = q_A(\hat{z}_a). \tag{3}$$

This projection step adapts the output of each frozen encoder and is subsequently trained to isolate cross-modal features. We then introduce two mask networks:

$$M_V, M_A : \mathbb{R}^{2d} \to [0, 1]^d, \tag{4}$$

each taking both $\tilde{z}_v$ and $\tilde{z}_a$ as input. Due to the semantic ambiguity and contextual diversity in audio-visual alignment, the relevant latent dimensions within the visual representation can vary depending on the specific context. For example, a single video clip may be paired with different types of audio, such as background music or voiceover narration, each requiring attention to distinct visual regions or semantic features. Accordingly, the masking function should be conditioned on both video and audio inputs. Conditioning on only one modality impairs the model's ability to disambiguate cross-modal variations, leading to suboptimal or unstable mask learning.

**Cross-Modal Reconstruction**   Let $S_V \subseteq [d]$ and $S_A \subseteq [d]$ represent the indices of dimensions selected by the soft masks. After applying a thresholding operation, we obtain binary supports $\tilde{S}_V \subseteq [d]$ and $\tilde{S}_A \subseteq [d]$, which indicate the dimensions where the mask value equals 1 (i.e., $\tilde{S}_V = \{i \in [d] \mid M_V(\tilde{z}_v, \tilde{z}_a)_i = 1\}, \tilde{S}_A = \{i \in [d] \mid M_A(\tilde{z}_v, \tilde{z}_a)_i = 1\}$). These binary masks are then used to construct the masked latent representations by retaining only the selected dimensions:

$$\tilde{z}_v = M_V(\tilde{z}_v, \tilde{z}_a) \odot \tilde{z}_v, \qquad \tilde{z}_a = M_A(\tilde{z}_v, \tilde{z}_a) \odot \tilde{z}_a, \tag{5}$$

and decode each using the corresponding latent diffusion decoders $g_A$ and $g_V$:

$$\hat{x}_a = g_A(\tilde{z}_v), \qquad \hat{x}_v = g_V(\tilde{z}_a). \tag{6}$$

To ensure that the masked latent remains informative, we reconstruct each modality from the masked latent of the other. This reconstruction objective acts as a constraint that prevents degenerate masking

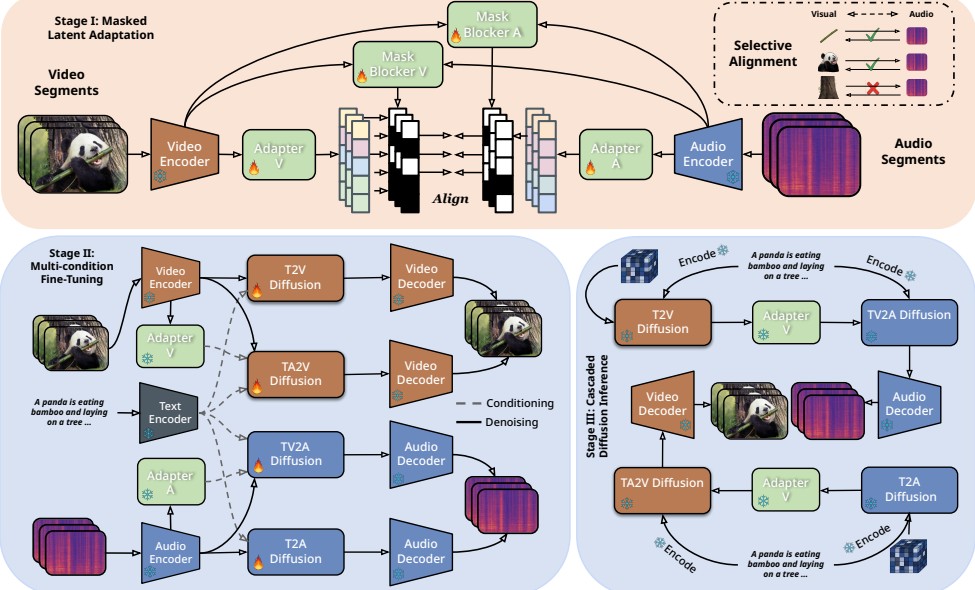

Figure 3: Overview of our proposed T2AV framework. The system involves training a learnable mask that selectively aligns the latents of each modality, filtering out irrelevant visual content (e.g., tree trunks) while preserving meaningful cues (e.g., bamboo being eaten). The aligned representations are then used to fine-tune the generator, adapting the multimodal conditions alongside the text condition, followed by generation through a latent diffusion model.

solutions. Without it, the sparsity loss alone would encourage all-zero masks, as they trivially minimize the L1 penalty while discarding all information [23, 50]. We reconstruct across modalities (i.e., audio from masked video latent and video from masked audio latent) rather than within the same modality. This cross-modal reconstruction forces the mask to preserve only the latent dimensions that are predictive of the other modality, thereby isolating the shared semantic structure. As a result, the model learns compact and meaningful latent supports that are truly cross-modally informative.

## 3.1 SAVA-Diffusion

In this section, we present the implementation of the proposed SAVA-Diffusion framework for Text-to-Audio-Video (T2AV) generation (See Figure 3). The framework consists of three stages: (1) a masked latent adaptation stage in which we train aligned projections of video-audio latents via selective masking, and (2) a fine-tuning stage that adapts the aligned latents as a joint condition alongside the text condition. (3) a cascaded diffusion generation stage, where high-fidelity audio and video outputs are synthesized using latent diffusion models in sequential orders.

**Stage I: Masked Latent Adaptation** In first stage, our method implements selective cross-modal alignment by learning to isolate the latent dimensions that are predictive of the other modality. We first obtain reparameterized embeddings $\tilde{z}_v$ and $\tilde{z}_a$ from the frozen encoders [53, 26] and adapters. These are passed to the modality-specific mask functions, each conditioned on both modalities, to produce binary masks that filter the latent features. The masked latents are then decoded to reconstruct the opposite modality. To encourage a consistent embedding geometry between modalities [51], we further include a direct alignment loss between $\tilde{z}_v$ and $\tilde{z}_a$ prior to masking. This stabilizes training and promotes representational coherence across modalities. The total objective is:

$$\mathcal{L}_{\text{total}} = \frac{1}{N} \sum_{i=1}^{N} \left[ \ell_A(x_a^{(i)}, g_A(M_V \odot \tilde{z}_v^{(i)})) + \ell_V(x_v^{(i)}, g_V(M_A \odot \tilde{z}_a^{(i)})) + \alpha \cdot \mathcal{L}_{\text{align}}(\tilde{z}_v^{(i)}, \tilde{z}_a^{(i)}) \right] \tag{7}$$
$$+ \lambda \left( \|M_V\|_1 + \|M_A\|_1 \right),$$

where $\ell_A$ and $\ell_V$ are cross-modal reconstruction losses, and the $\ell_1$ regularization encourages sparsity in the learned supports. $\mathcal{L}_{\text{align}}$ measures the distance (e.g., normalized $\ell_2$) between the unmasked latent representations, and $\alpha$ controls the alignment strength.

**Stage II: Multi-condition Fine-tuning**   In Stage II, the TV2A and TA2V diffusion backbones are fine-tuned separately, each to use the cross-modal latent learned in Stage I, while keeping the decoders $g_V, g_A$ frozen. For audio, given the text embedding $z_t = f_t(x_t)$ and the aligned visual latent $\tilde{z}_v = q_V(z_v^T)$, we form $c_A = [z_t; \phi_A(\tilde{z}_v)]$ and adapt the conditioning pathway via LoRA [12], similarly, for video we form $c_V = [z_t; \phi_V(\tilde{z}_a)]$ from the aligned audio latent $\tilde{z}_a = q_A(z_a^T)$, the objective is the diffusion loss:

$$\mathcal{L}_{\text{FT-A}} = \mathbb{E}_{x_a, \epsilon, t} \left\| \epsilon - \epsilon_{\theta_A}\big(x_a^{(t)}, t, c_A\big) \right\|_2^2 \qquad \mathcal{L}_{\text{FT-V}} = \mathbb{E}_{x_v, \epsilon, t} \left\| \epsilon - \epsilon_{\theta_V}\big(x_v^{(t)}, t, c_V\big) \right\|_2^2. \tag{8}$$

with LoRA parameterization $W' = W + BA$ on selected cross-attention or FiLM layers. The total objective is not coupled during optimization, instead, we run distinct trainings. LoRA only on conditioning layers (mid-block and a few down/up blocks), and all backbone convolutions and decoders frozen to preserve pretrained priors while teaching each model, in isolation, to respond to its new cross-modal condition. In addition, we fine-tune the individual T2A and T2V models to further enhance generation quality across modalities for the inference pipeline.

**Stage III: Cascaded Diffusion Inference**   Building on the aligned latent representations from stage I and fine-tuned diffusion models from stage II, we generate video and audio in a cascaded manner using independently finetuned single-modal diffusion models (T2A, T2V) [53, 26] and multi-model diffusion models (TA2V, TV2A). As illustrated in Figure 3, the process begins by generating a video from a text prompt using a T2V diffusion model. The resulting visual latent $z_v^T$ is then adapted through a lightweight projection network $\mathcal{P}_\theta$, producing an audio-guiding latent $\tilde{z}_a$ that encodes visually grounded cues. This latent conditions the subsequent audio generation, serves as a supervision signal to finetune the audio diffusion model for improved semantic coherence. Similarly, we can generate video conditioned on both audio and text latents. By structuring the process in this cascaded fashion, we ensure that the audio is aligned with the generated visual/audible content. The detailed formulation of the reverse diffusion process for both modalities is provided in Appendix C. Notably, the pretrained diffusion encoders remain frozen during Stage I, and only the adapters are updated; fine-tuning of the diffusion models is performed in Stage II to enhance generation quality while maintaining modularity and efficiency.

## 4   Theoretical Analysis

In this section we show that our masked cross-modal reconstruction with an $\ell_1$-penalty provably recovers exactly the shared latent factors between video and audio, i.e. the minimal Markov blankets on the pretrained features, even when those features are entangled. By faithfulness and d-separation on the latent DAG [35] over $(Z_V, Z_A, X_V, X_A)$, there exist unique index sets $S_V^\dagger \subseteq [d]$, $S_A^\dagger \subseteq [d]$, which are the minimal Markov blankets of $X_A$ in $Z_V$ and of $X_V$ in $Z_A$, respectively. Equivalently, $S_V^\dagger$ is the smallest subset satisfying that conditioning on $\{Z_{V,i} : i \in S_V^\dagger\}$ renders all other latent coordinates irrelevant to $X_A$. The analogous property holds for $S_A^\dagger$.

To make precise what it means for two sets of latent factors to capture all and only the shared information, we introduce the following definition.

**Definition 1** (Minimum Sufficient Latents). *Given index sets $\tilde{S}_V, \tilde{S}_A \subseteq [d]$, we say that the pairs $\big(\tilde{Z}_V^{\tilde{S}_V}, \tilde{Z}_A^{\tilde{S}_A}\big)$ are* Minimum Sufficient Latents *if they satisfy*

$$I\big(\tilde{Z}_V^{\tilde{S}_V}; X_A\big) = I\big(Z_V^{S_V^\dagger}; X_A\big), \ I\big(\tilde{Z}_{V_j}; X_A \mid \tilde{Z}_V^{\tilde{S}_V}\big) = 0 \ \forall j \notin \tilde{S}_V,$$

$$I\big(\tilde{Z}_A^{\tilde{S}_A}; X_V\big) = I\big(Z_A^{S_A^\dagger}; X_V\big), \ I\big(\tilde{Z}_{A_j}; X_V \mid \tilde{Z}_A^{\tilde{S}_A}\big) = 0 \ \forall j \notin \tilde{S}_A.$$

**Key Assumptions**   We require four conditions (see Appendix B for formal definitions):

1. **(DAG & d-Separation)** There is a latent DAG over $(Z_V, Z_A, X_V, X_A)$ whose minimal Markov blankets $S_V^\dagger, S_A^\dagger$ correspond to the truly shared factors.

2. **(Block-wise Reparameterization)** The class of invertible maps $q_V, q_A$ is rich enough that there exists a reparameterization under which the shared block $S_V^\dagger$ (resp. $S_A^\dagger$) becomes an axis-aligned subset $\tilde{S}_V^\dagger$ (resp. $\tilde{S}_A^\dagger$) of the coordinates.

3. **(Decoder Universality)** The decoder families $Q_{g_A}, Q_{g_V}$ can approximate any conditional distribution, so that minimizing cross-entropy is equivalent to minimizing true conditional entropy.

4. **(Mask Universality & Penalty-Range)** The masks can implement any support selection per example, and the sparsity weight $\lambda$ lies strictly between the smallest shared-factor contribution and the largest non-shared contribution (see Assumption 4).

Above assumptions are commonly used. First, a latent-variable DAG with faithfulness (Assumption 1) underlies most generative models in vision and audio, and the Markov blanket then exactly characterizes the shared information. This is the fundamental assumption in Causality [35]. Second, block-wise reparameterization (Assumption 2) merely requires that our invertible networks $q_V, q_A$ have sufficient capacity to "whiten" or disentangle the small block of truly shared latents; in practice modern normalizing-flow and invertible-residual architectures easily satisfy this. Third, decoder universality (Assumption 3) is standard in representation learning deep decoders with enough width and nonlinearity can approximate any conditional density arbitrarily well, so cross-entropy minimization recovers true conditional entropy. Mask universality implies stipulate that our mask networks are expressive enough to pick any subset of coordinates per example. All these universality have been supported by universal approximation theory of deep learning methods [14]. Finally, penalty-range requirement (Assumptions 4) implies that the sparsity weight $\lambda$ can be chosen (e.g. via cross-validation) to lie between the minimal utility of a shared factor and the maximal spurious contribution of a non-shared factor. In practice, we can just make $\lambda$ be sufficiently small. Together, these common assumptions ensure our theoretical guarantees apply to many practical architectures.

**Lemma 1** (Sufficientness of Reconstruction). *Fix any invertible $q_V$. Under Assumptions 1-4, any mask–decoder pair $(M_V, g_A)$ that minimizes $\mathbb{E}[-\log Q_{g_A}(X_A \mid M_V \odot \tilde{Z}_V)]$ must satisfy, for every example, $I\big(\tilde{Z}_V^{S_V(\tilde{Z}_V, \tilde{Z}_A)}; X_A\big) = I\big(\tilde{Z}_V; X_A\big)$. In other words, the selected coordinates form a sufficient statistic for $X_A$.*

This lemma shows that if we only optimize the reconstruction loss (cross-entropy) then the learned mask necessarily keeps *all* the information in $\tilde{Z}_V$ that is relevant to predicting $X_A$. In other words, the selected subset of coordinates forms a sufficient statistic for the audio modality, capturing every bit of shared information from the video embedding.

**Lemma 2** (Sparsity-Induced Minimality). *Fix any invertible $q_V$. Under Assumptions 3-4, the joint minimizer $(M_V^*, g_A^*) = \arg\min_{M_V, g_A} \big\{ \mathbb{E}\big[-\log Q_{g_A}(X_A \mid M_V \odot \tilde{Z}_V)\big] + \lambda \mathbb{E}[\|M_V\|_1] \big\}$ satisfies, for every example,*

$$S_V^*(\tilde{Z}_V, \tilde{Z}_A) = \tilde{S}_V^{\dagger}, \ I\big(\tilde{Z}_{V,j}; X_A \mid \tilde{Z}_V^{\tilde{S}_V^{\dagger}}\big) = 0 \ \forall j \notin \tilde{S}_V^{\dagger}. \tag{9}$$

*That is, the mask prunes away non-shared coordinates, recovering exactly the minimal shared block.*

This lemma establishes that once we add a sparsity penalty on the mask, the model discards every coordinate that does not uniquely contribute to cross-modal reconstruction. The result is the *minimal* subset of features which are precisely the shared latent block. Therefore no redundant or modality-specific information remains.

**Theorem 1** (Global Block-Alignment and Recovery). *Under Assumptions 1, 2, 3, and 4 the global minimizer of Objective 7 yields $\big(\tilde{Z}_V^{\tilde{S}_V^*}, \tilde{Z}_A^{\tilde{S}_A^*}\big)$ that satisfies Definition 2.*

This theorem combines the sufficiency and minimality results in both directions (i.e., video→audio and audio→video) and shows that our simple mask-and-reconstruct framework provably extracts exactly the shared latent variables and eliminates all modality-specific components.

# 5 Experiments

## 5.1 Experiment Setup

**Dataset** We conduct experiments on two benchmark datasets: VGGSound [5] and AudioCaps [22]. VGGSound comprises approximately 200K 10-second video clips spanning 310 sound classes, with strong audio-visual correspondence ensured by the presence of visible sound sources. Following the protocol in [52], we sample 5k and 3K clips from the train and test split, respectively, and annotate

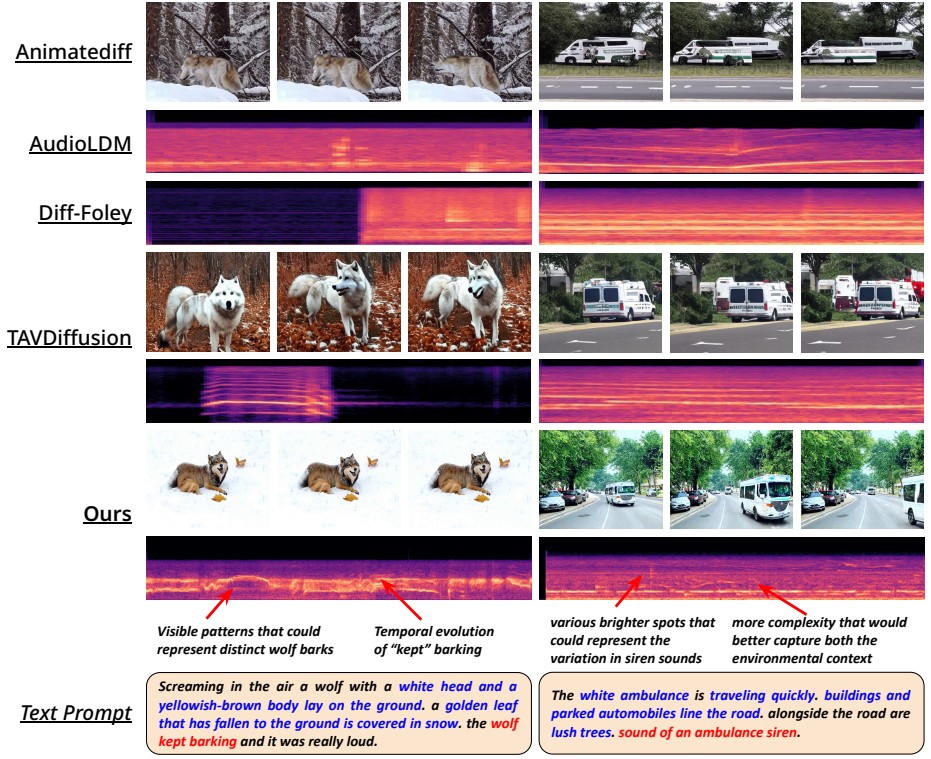

Figure 4: Text-to-Audio-Video generation results. We use the same text prompt as in [33] for our demonstration and compare our method against multiple baselines (Animatediff [8], AudioLDM [26], Diff-Foley [31], and TAVDiffusion [33]). Compared to prior methods, our approach (unidirectional setting as illustrated) produces higher quality and aligned video and audio content.

| Method | VGGSound+ | | | | AudioCaps | | | |
|---|---|---|---|---|---|---|---|---|
| | FVD ↓ | FAD ↓ | AVHScore ↑ | CAVPSIM ↑ | FVD ↓ | FAD ↓ | AVHScore ↑ | CAVPSIM ↑ |
| Two-Streams | 768.5 | 6.29 | 0.058 | 0.104 | 961.4 | 7.36 | 0.041 | 0.165 |
| CasC-Diff | 768.5 | 7.53 | 0.144 | 0.126 | 961.4 | 9.51 | 0.092 | 0.192 |
| TAVDiff [33] | 956.3 | 8.94 | 0.162 | 0.098 | 1131.9 | 8.43 | 0.105 | 0.182 |
| CoDi [41] | 709.4 | 8.36 | 0.108 | 0.149 | 902.5 | 9.07 | 0.098 | 0.211 |
| JavisDiT [29] | 697.4 | 6.17 | 0.153 | 0.140 | 801.2 | 7.55 | 0.104 | 0.207 |
| Unidirectional | **662.9** | **5.49** | 0.206 | 0.165 | 817.6 | **7.32** | 0.142 | 0.230 |
| Bidirectional | 701.4 | - | **0.217** | **0.183** | 852.4 | - | **0.157** | **0.242** |

Table 1: Quantitative comparison. Our method outperforms existing baselines in both generative quality metrics and alignment metrics, demonstrating improvements in fidelity as well as cross-modal consistency. For the unidirectional setting, we directly adopt the fine-tuned T2V model for video generation. The generated audio for both the bidirectional and unidirectional settings is identical.

them with text prompts using VideoBlip [54], as adopted in [33]. AudioCaps consists of 46K audio clips paired with human-written captions sourced from AudioSet, and serves as a standard benchmark for audio-language grounding. We also sample 5K paired clips from the training split. To facilitate alignment learning and fine-tuning, we merge the training sets of both datasets, and perform evaluation separately on each test set.

**Implementation Details** To adapt the diffusion models to the target data domains, we first fine-tune the video and audio diffusion models independently using the training set, respectively. For video generation, we employ the pretrained CogVideoX1.5 [53], and extract latent representations using its VAE encoder. For audio, we adopt AudioLDM [26], which integrates a pretrained CLAP encoder [49] for audio feature extraction. The latent dimensionality of aligned embeddings for audio generation is fixed at 512. Each generated sample has a duration of 10 seconds, with video rendered at 16 frames per second and audio sampled at 48 kHz. Our adapter and masking modules are implemented as multilayer perceptrons. For the masking mechanism, we evaluate both soft masks (sigmoid outputs as weights) and hard masks, obtained by thresholding at 0.5. The loss weights $\lambda_1$ and $\lambda_2$ are empirically set to 5 and 0.1, respectively.

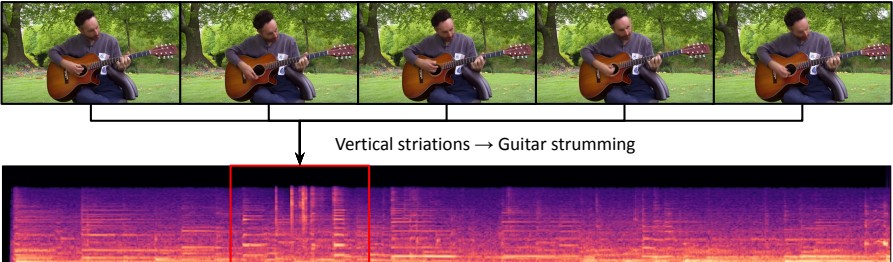

Text prompt: A man playing guitar in the park.

Figure 5: Temporal alignment between visual motion and acoustic patterns. The strumming motion of the guitarist's hand aligns with vertical striations in the spectrogram, indicating synchronized transient audio events.

**Evaluation Metrics** We assess perceptual quality of the generated video and audio using Fréchet Video Distance (FVD) [43] and Fréchet Audio Distance (FAD) [37], respectively. Cross-modal semantic alignment is measured by AVHScore [33], while CAVP similarity [31] evaluates temporal synchronization. For V2A performance, we adopt the evaluation protocol from [52], including KL divergence, Inception Score (ISc), Fréchet Distance (FD), and FAD.

**Baselines** We compare our method against two state-of-the-art T2AV approaches: TAVDiffusion [33] and CoDi [41], using the same text prompts for all models. Additionally, we include: (1) a Cascaded pipeline that uses Animatediff [8] for video generation followed by V2A-Mapper [45] for audio synthesis, and (2) a Two-Stream approach in which video and audio are independently generated from the same text prompt using Animatediff and AudioLDM. For V2A generation, we also compare with the contrastive alignment method in [52] and SpecVQGAN [20], a spectrogram-based audio generator employing vector quantization.

## 5.2 Main Results

**T2AV Generative Quality** Table 1 presents the quantitative comparison of our proposed method against existing T2AV generation baselines on the text-labeled VGGSound and AudioCaps datasets. Our method consistently achieves the best performance across all reported metrics. On VGGSound, it reduces FVD and FAD to 662.9 and 5.49, respectively, reflecting significant improvements in both video and audio generation fidelity. Compared to the best-performing baseline [41], our method achieve a relative reduction of 6.5% in FVD and 16.0% in FAD. In terms of semantic alignment, our model achieves the highest AVHScore of 0.206 and 0.142 on VGGSound and AudioCaps, respectively, demonstrating improved correspondence between generated content and the input descriptions. A similar trend is observed in the qualitative results shown in Figure 4. For example, in the wolf scenario, our generated video better captures key semantic attributes such as the white head and yellow-brown body, while the visual background and ambient objects more faithfully reflect the textual prompt. Likewise, the ambulance scene displays correct object types, vehicle motion, and contextual elements like roadside greenery and traffic, showing high semantic fidelity across modalities.

**Semantic and Temporal Alignment** As shown in Table 1, our method achieves the highest CAVP-SIM scores on both VGGSound and AudioCaps, indicating improvement on cross-modal temporal alignment. This is further illustrated in Figure 4, where the temporal evolution of audio patterns (e.g., barking or sirens) closely corresponds with visual events. In the wolf example, distinct spectrogram patterns align with repeated barking motions, while the ambulance scenario shows dynamic spectrogram textures matching siren intensity and vehicle motion. To further highlight this property, Figure 5 visualizes a man strumming a guitar, where the rhythmic hand motion aligns with vertically striated spectrogram features indicative of transient guitar strokes. These results collectively confirm that our method not only generates high-quality content but also preserves temporal synchronization across modalities. Additional examples are provided in Appendix D.

**V2A Generation** To further evaluate the effectiveness of cross-model alignment, we assess the video-to-audio (V2A) generation performance using a subcomponent of our model. As shown in Table 2, our approach outperforms existing V2A baselines: SpecVQGAN [20] and SeeHear [52] across most metrics, achieving better KL, ISc and FAD. These results indicate that our model effectively captures the shared semantic and temporal information between video and audio, enabling high-quality cross-

| Method | KL↓ | ISc↑ | FD↓ | FAD↓ |
|---|---|---|---|---|
| SpecVQGAN [20] | 3.290 | 5.108 | 37.269 | 7.736 |
| SeeHear-Vani [52] | 3.203 | 5.625 | 40.457 | 6.850 |
| SeeHear-Full [52] | 2.619 | 5.831 | **32.920** | 7.316 |
| Ours | **2.128** | **5.677** | 39.534 | **6.155** |

Table 2: Video-to-Audio Generation Results. Our method outperforms existing V2A baselines across most evaluation metrics, demonstrating noticeable improvements in audio fidelity.

| Mask | FVD ↓ | FAD ↓ | AVHScore ↑ | CAVPSIM ↑ |
|---|---|---|---|---|
| □ | 662.9 | 6.95 | 0.175 | 0.141 |
| ○ | 662.9 | 6.08 | 0.192 | 0.144 |
| △ | 662.9 | 5.49 | 0.206 | 0.165 |

Table 3: Ablation study on masking input modalities. □: no masking, direct alignment, ○: only takes video modality embeddings as the input, △: takes both video and audio modality embeddings as the input.

modal generation. The performance of this subcomponent further validates the robustness of our alignment strategy.

## 5.3 Ablation Study

**Study on Mask Input** We conduct an ablation study to assess the impact of different mask input configurations on cross-modal generation quality in Table 3. We observe that when no masking is applied (□), performance is significantly lower across all metrics, indicating that direct alignment without filtering introduces noise and misalignment. Conditioning the mask on video alone (○) yields moderate improvements, suggesting that video features contain partial cues for predicting shared content. However, the best performance is achieved when the mask is conditioned on both video and audio embeddings (△), resulting in the lowest FAD and highest AVHScore and CAVPSIM. This confirms our hypothesis that observing both modalities enables the mask to more accurately isolate cross-modally relevant dimensions, thereby enhancing semantic consistency and temporal alignment in the generated outputs.

**Effect of Temporal Segmentation** We conduct an ablation study to investigate the impact of different temporal segmentation strategies on the performance of T2AV generation. Specifically, given a 10-second video/audio clip, we divide the content into sub-clips of varying lengths (ranging from 1s to 10s) and use the aligned segments for fine-tuning the pretrained diffusion models and training

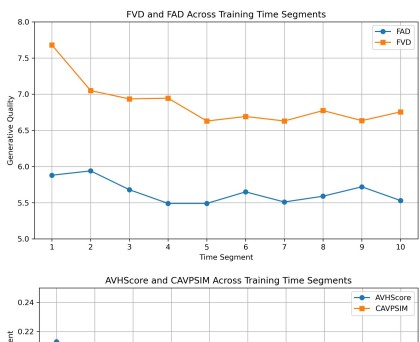

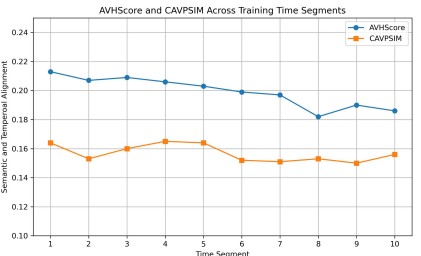

Figure 6: Ablation on different time segment lengths. We find that longer segments improve generative quality, while shorter segments benefit alignment .

the alignment modules, including the masking functions and adapters. As illustrated in Figure 6, longer segment durations consistently improve generative quality, as measured by FAD and FVD, likely due to providing richer temporal context for fine-tuning the diffusion backbones. In contrast, shorter segments yield stronger performance in alignment metrics such as AVHScore and CAVPSIM, suggesting that temporally concise segments reduce misalignment and noise during cross-modal training. Based on this trade-off, we select a 5-second segment length as a balanced choice that supports both high generative fidelity and accurate audio-visual alignment.

## 6 Conclusion

We presented a multi-stage framework for text-to-audio-video (T2AV) generation that addresses the challenge of semantic and temporal misalignment between modalities. Our method introduces a masked latent adaptation mechanism that selectively aligns video representations with audio embeddings using a learnable adapter and relevance mask. During inference, we leverage a cascaded diffusion structure in which video is generated from text, and audio is subsequently synthesized conditioned on both text and the adapted video latent. This design ensures coherence across modalities while maintaining flexibility by reusing single-modal diffusion models. Extensive experiments demonstrate that our approach improves cross-modal consistency and achieves state-of-the-art results on multimodal generation benchmarks.

# 7 Acknowledgement

The authors would like to thank Suqin Yuan and Muyang Li for their valuable feedback during the project. Jiyang Zheng is supported by the CSIRO Next Generation Graduates and AI for Missions PhD program. Tongliang Liu is partially supported by the following Australian Research Council projects: FT220100318, DP220102121, LP220100527, LP220200949.

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

## Appendix

## A   Related Works

**Text-to-Audio-Video Generation**   Text-to-Audio-Video (T2AV) generation aims to synthesize audio and video streams that are semantically and temporally aligned, conditioned on a single text

prompt. The task extends beyond text-to-video (T2V) and text-to-audio (T2A) generation by requiring consistency across modalities. Recent advances in T2V [4, 8, 21, 47, 55, 25] and T2A [1, 17, 27, 32, 40] have enabled high-quality content generation in each modality. However, generating them independently often results in misaligned outputs, as the modalities are not conditioned on each other. A simple alternative is a cascaded approach, where one modality (e.g., video) is generated first and used to condition the other (e.g., audio). While this improves synchronization, it may propagate errors and lead to inconsistencies with the original text. To address these issues, recent T2AV methods propose joint modeling strategies. CoDi [41] unifies generation across multiple modalities in a single diffusion framework via aligning prompt encoders (text, image, video, audio) into a shared input space using contrastive learning, with text as the central bridging modality. [52] aligns pretrained T2V and T2A models via a shared semantic space using ImageBind. TAVDiffusion [33] adopts a two-stream latent diffusion model and addresses alignment via cross-attention and contrastive learning. Nevertheless, joint modeling of audio and visual modalities requires careful alignment of representations to preserve both semantic consistency and temporal synchronization. Our framework complements existing approaches by introducing a targeted alignment mechanism that mitigates the impact of noisy or partial correspondences, leading to more faithful and consistent T2AV generation.

**Cross-Modal Alignment** Cross-modal alignment is crucial for integrating information from different modalities, facilitating tasks such as retrieval, recognition [56, 9], and generation [19, 58, 57, 46]. The goal is to project modality-specific features into a shared embedding space where semantically related inputs are closely aligned. In the vision-language domain [18, 15, 16, 30], CLIP [36] has become a standard framework, while CLAP [49] and CAVP [31] extend contrastive alignment to audio-language and vision-audio pairs, respectively. ImageBind [7] further generalizes this approach to unify multiple modalities in a single embedding space. Such alignment modules are integral to conditional generative models [31]. While early approaches trained modality encoders from scratch, recent work shows that frozen foundation models can be effectively adapted using lightweight projectors [11, 34]. In video-to-audio generation, V2A-Mapper [45] learns a projection from CLIP to CLAP features using a simple MLP, enabling audio generation conditioned on vision without retraining large-scale models. Despite these advances, aligning the correct semantic content across modalities remains challenging. Representations often entangle modality-specific and irrelevant information, leading to noisy alignment. SmartCLIP [51] identifies this issue in vision-language models, showing that CLIP embeddings often entangle unrelated concepts due to coarse-grained alignment. These findings underscore a broader challenge in multimodal generation: how to align information across modalities such that the learned representations do not introduce inconsistencies in the generated outputs. Our work addresses this by introducing a masked adapter module that enables efficient and selective alignment between pretrained modality-specific encoders. By focusing alignment on semantically relevant regions, our method mitigates noisy correspondence and improves consistency in T2AV generations.

# B  Theoretical Results and Proofs

## B.1  Notations and Definitions

We introduce key notations and definitions as follows.

Let $X_V \in \mathcal{X}_V$ (video) and $X_A \in \mathcal{X}_A$ (audio) be two modalities. Pretrained encoders produce

$$\hat{Z}_V = (\hat{Z}_{V,1}, \ldots, \hat{Z}_{V,d}), \quad \hat{Z}_A = (\hat{Z}_{A,1}, \ldots, \hat{Z}_{A,d}) \in \mathbb{R}^d. \tag{10}$$

We introduce learnable invertible reparameterizations

$$q_V, q_A : \mathbb{R}^d \to \mathbb{R}^d, \tag{11}$$

and define

$$\tilde{Z}_V = q_V(\hat{Z}_V), \quad \tilde{Z}_A = q_A(\hat{Z}_A). \tag{12}$$

We learn *mask functions*

$$M_V : \mathbb{R}^d \times \mathbb{R}^d \to \{0,1\}^d, \quad M_A : \mathbb{R}^d \times \mathbb{R}^d \to \{0,1\}^d, \tag{13}$$

so that for each sample the model produces binary masks

$$M_V(\tilde{Z}_V, \tilde{Z}_A), \ M_A(\tilde{Z}_V, \tilde{Z}_A) \in \{0,1\}^d, \tag{14}$$

Denote their supports (or selected indices) by

$$\tilde{S}_V := S_V(\tilde{Z}_V, \tilde{Z}_A) = \{\, i : M_{V,i}(\tilde{Z}_V, \tilde{Z}_A) = 1\,\}, \tag{15}$$

$$\tilde{S}_A := S_A(\tilde{Z}_V, \tilde{Z}_A) = \{\, i : M_{A,i}(\tilde{Z}_V, \tilde{Z}_A) = 1\,\}. \tag{16}$$

Let $\tilde{Z}_V^{\tilde{S}_V} \subseteq \tilde{Z}_V$ and $\tilde{Z}_A^{\tilde{S}_A} \subseteq \tilde{Z}_A$ be the selected variables induced by learnable masks, i.e.,

$$\tilde{Z}_V^{\tilde{S}_V} = (\tilde{Z}_{V,i})_{i \in \tilde{S}_V}, \quad \tilde{Z}_A^{\tilde{S}_A} = (\tilde{Z}_{A,i})_{i \in \tilde{S}_A}, \quad \overline{\tilde{S}_V} = [d] \setminus \tilde{S}_V \text{ and } \overline{\tilde{S}_A} = [d] \setminus \tilde{S}_A, \tag{17}$$

for any set $\tilde{S}_V \subseteq [d]$ and set $\tilde{S}_A \subseteq [d]$.

Write $S_V^\dagger \subseteq [d]$ (resp. $S_A^\dagger$) for the *true* minimal Markov blanket of $X_A$ in $Z_V$ (resp. of $X_V$ in $Z_A$). In other words, $S_V^\dagger$ is the smallest subset satisfying that conditioning on $\{Z_{V,i} : i \in S_V^\dagger\}$ renders all other latent coordinates irrelevant to $X_A$. The analogous property holds for $S_A^\dagger$.

Reconstruction quality is measured via true conditional entropy:

$$H(X_A \mid \tilde{Z}_V^{\tilde{S}_V}) = -\mathbb{E}\big[\log p(X_A \mid \tilde{Z}_V^{\tilde{S}_V})\big], \quad H(X_V \mid \tilde{Z}_A^{\tilde{S}_A}) = -\mathbb{E}\big[\log p(X_V \mid \tilde{Z}_A^{\tilde{S}_A})\big]. \tag{18}$$

Let $Q_{g_A}(X_A \mid \cdot)$ and $Q_{g_V}(X_V \mid \cdot)$ be decoder families. We optimize

$$\mathcal{L}_V(M_V, q_V, g_A) = \mathbb{E}\Big[-\log Q_{g_A}\big(X_A \mid M_V(\tilde{Z}_V, \tilde{Z}_A) \odot \tilde{Z}_V\big)\Big] + \lambda\, \mathbb{E}\big[\|M_V(\tilde{Z}_V, \tilde{Z}_A)\|_1\big], \tag{19}$$

and symmetrically $\mathcal{L}_A(g_V, q_V, M_A)$ for audio→video.

**Definition 2** (Minimum Sufficient Latents). *Given index sets $\tilde{S}_V, \tilde{S}_A \subseteq [d]$, we say that the pairs $\big(\tilde{Z}_V^{\tilde{S}_V}, \tilde{Z}_A^{\tilde{S}_A}\big)$ are* Minimum Sufficient Latents *if they satisfy*

$$I\big(\tilde{Z}_V^{\tilde{S}_V}; X_A\big) = I\big(Z_V^{S_V^\dagger}; X_A\big),\ I\big(\tilde{Z}_{V_j}; X_A \mid \tilde{Z}_V^{\tilde{S}_V}\big) = 0 \ \ \forall j \notin \tilde{S}_V,$$

$$I\big(\tilde{Z}_A^{\tilde{S}_A}; X_V\big) = I\big(Z_A^{S_A^\dagger}; X_V\big),\ I\big(\tilde{Z}_{A_j}; X_V \mid \tilde{Z}_A^{\tilde{S}_A}\big) = 0 \ \ \forall j \notin \tilde{S}_A.$$

## B.2 Assumptions

We introduce the assumptions required by our method as follows.

**Assumption 1** (DAG & d-Separation). *The joint distribution of $(Z_V, Z_A, X_V, X_A)$ factors according to a DAG satisfying the global Markov property and faithfulness.*

*Hence for any $S_V \subseteq [d]$,*

$$X_A \perp Z_V^{\overline{S}_V} \mid Z_V^{S_V} \quad \Longleftrightarrow \quad I\big(Z_{V,i}; X_A \mid Z_V^{S_V}\big) = 0 \ \forall i \notin S_V, \tag{20}$$

*and the same condition holds when $V$ and $A$ are interchanged.*

**Assumption 2** (Block-wise Reparameterization). *The joint function class for $(q_V, q_A)$ is rich enough that there exist invertible maps*

$$q_V^*, q_A^* : \mathbb{R}^d \to \mathbb{R}^d \tag{21}$$

*and index-sets $\tilde{S}_V^\dagger, \tilde{S}_A^\dagger \subseteq [d]$ satisfying*

$$I\big(q_V^*(Z_V)^{\tilde{S}_V^\dagger}; X_A\big) = I\big(Z_V^{S_V^\dagger}; X_A\big), \quad I\big(q_V^*(Z_V)_j; X_A \mid q_V^*(Z_V)^{\tilde{S}_V^\dagger}\big) = 0 \ \ \forall j \notin \tilde{S}_V^\dagger. \tag{22}$$

**Assumption 3** (Decoder Universality). *For any $S \subseteq [d]$, $\min_{g_A} \mathbb{E}[-\log Q_{g_A}(X_A \mid \tilde{Z}_V^{\tilde{S}_V})] \to H(X_A \mid \tilde{Z}_V^{\tilde{S}_V})$ and similarly for $X_V \mid \tilde{Z}_A^S$.*

**Assumption 4** (Mask Universality). *The mask networks $M_V, M_A$ are sufficiently expressive to realize any mapping $\mathbb{R}^d \times \mathbb{R}^d \to \{0,1\}^d$, i.e. choose any support $S \subseteq [d]$ for each sample.*

**Assumption 5** (Penalty-Range). *For any subset $\tilde{S}_V \subseteq [d]$ and index $i \in [d]$, define*

$$\Delta_{V,i}(\tilde{S}_V) = I\big(\tilde{Z}_{V,i}; X_A \mid \tilde{Z}_V^{(\tilde{S}_V \setminus \{i\})}\big),$$

*which is the mutual information between $\tilde{Z}_{V,i}$ and $X_A$ conditioned on the remaining variables $\tilde{Z}_V^{(\tilde{S}_V \setminus \{i\})} = \{\tilde{Z}_{V,j} : j \in \tilde{S}_V \setminus \{i\}\}$. There exists a constant $\lambda$ such that*

$$\max_{j \notin \tilde{S}_V^\dagger} \Delta_{V,j}([d]) \ < \ \lambda \ < \ \min_{i \in \tilde{S}_V^\dagger} \Delta_{V,i}(\tilde{S}_V^\dagger),$$

*and the same condition holds when $V$ and $A$ are interchanged.*

Note that above assumptions are common. Assumption 1 is a fundamental assumption in causality [35]. Assumption 2 merely requires that our networks $q_V, q_A$ have sufficient capacity to "whiten" or disentangle the small block of truly shared latents. Assumption 3 assumes that deep decoders can approximate any conditional density arbitrarily well, so cross-entropy minimization recovers true conditional entropy. Assumption 4 implies stipulate that our mask networks are expressive enough to pick any subset of coordinates per example. All these assumptions 2, 3, 4 have been supported by universal approximation theory of deep learning methods [14]. Finally, Assumptions 5 implies that the sparsity weight $\lambda$ can be chosen to lie between the minimal utility of a shared factor and the maximal spurious contribution of a non-shared factor. In practice, we can just make $\lambda$ be sufficiently small.

## B.3 Theoretical Results

The following lemma shows that, by minimizing the cross-entropy loss of a decoder trained to reconstruct a short audio segments from the selected learned representations of video frames, one asymptotically recovers the conditional entropy of reconstructed short audio segments given those representations. The same result holds when swapping the roles of video $V$ and the audio $A$.

**Lemma 3** (Cross-Entropy Reduction to Conditional Entropy). *Under Assumption 3, for any fixed mask function $M_V$ and fixed $q_V$, we have*

$$\min_{g_A} \mathbb{E}\Big[-\log Q_{g_A}\big(X_A \mid M_V(\tilde{Z}_V, \tilde{Z}_A) \odot \tilde{Z}_V\big)\Big] \longrightarrow \mathbb{E}\Big[H\big(X_A \mid \tilde{Z}_V^{\tilde{S}_V}\big)\Big], \tag{23}$$

*where $\tilde{Z}_V = q_V(Z_V)$ and $S_V(\tilde{z}_V, \tilde{z}_A) = support\big(M_V(\tilde{z}_V, \tilde{z}_A)\big)$.*

*Proof.* Let

$$L(M_V, g_A) = \mathbb{E}\Big[-\log Q_{g_A}\big(X_A \mid M_V(\tilde{Z}_V, \tilde{Z}_A) \odot \tilde{Z}_V\big)\Big]. \tag{24}$$

By the interchange of minima,

$$\min_{M_V, g_A} L(M_V, g_A) = \min_{M_V}\Big[\min_{g_A} L(M_V, g_A)\Big]. \tag{25}$$

Fix any mask $M_V$. Then by Assumption 3 (Decoder Universality),

$$\min_{g_A} L(M_V, g_A) = \min_{g_A} \mathbb{E}\big[-\log Q_{g_A}(X_A \mid \tilde{Z}_V^{\tilde{S}_V})\big] \longrightarrow \mathbb{E}\big[H(X_A \mid \tilde{Z}_V^{\tilde{S}_V})\big]. \tag{26}$$

Note that $(X_A \mid M_V(\tilde{Z}_V, \tilde{Z}_A) \odot \tilde{Z}_V = Z_V^{\tilde{S}_V}$ by definition, since the mask selects exactly those components. Therefore, the proof is complete. $\square$

The following lemma shows that any mask–decoder pair minimizing the cross-entropy reconstruction loss inevitably selects a subset of video representations that retains the full mutual information with the audio segment, i.e., it forms a sufficient statistic for the audio segment. The same result holds when swapping the roles of video $V$ and the audio $A$.

**Lemma 4** (Sufficientness of Reconstruction). *Fix any invertible $q_V$. Under Assumptions 1–4, any mask–decoder pair $(M_V, g_A)$ that minimizes $\mathbb{E}[-\log Q_{g_A}(X_A \mid M_V \odot \tilde{Z}_V)]$ must satisfy, for every sample,*

$$I\big(\tilde{Z}_V^{\tilde{S}_V}; X_A\big) = I\big(\tilde{Z}_V; X_A\big). \tag{27}$$

*In other words, the selected coordinates form a sufficient statistic for $X_A$.*

*Proof.* For notational simplicity, we omit the arguments $(\tilde{Z}_V, \tilde{Z}_A)$ when writing $\hat{M}_V(\tilde{Z}_V, \tilde{Z}_A)$. Fix $q_V$ and consider any minimizer $(\hat{M}_V, \hat{g}_A)$ of the cross-entropy. By Lemma 3, this pair also minimizes $\mathbb{E}\big[H(X_A \mid \tilde{Z}_V^{\tilde{S}_V})\big]$. Under Assumption 4, the mask $M_V$ is expressive enough to choose the index set $\tilde{S}_V$ arbitrarily for each sample. Hence the minimization decomposes per sample: for each $(\tilde{z}_V, \tilde{z}_A)$, we pick

$$S_V(\tilde{z}_V, \tilde{z}_A) \in \arg\min_{\tilde{S} \subseteq [d]} H\big(X_A \mid \tilde{Z}_V^{\tilde{S}_V} = \tilde{z}_V^{\tilde{S}_V}\big). \tag{28}$$

Recall that for any fixed $\tilde{S}_v$,

$$H(X_A \mid \tilde{Z}_V^{\tilde{S}_V}) \;=\; H(X_A) - I(\tilde{Z}_V^{\tilde{S}_V}; X_A), \tag{29}$$

By the causal faithfulness and causal Markov properties [35] (analogous to Assumption 1, but applied to the variables produced by the neural network), it gives

$$I(\tilde{Z}_V^{\tilde{S}_V}; X_A) \leq I(\tilde{Z}_V; X_A) \quad \Longleftrightarrow \quad H(X_A \mid \tilde{Z}_V^{\tilde{S}_V}) \geq H(X_A \mid \tilde{Z}_V). \tag{30}$$

Hence the unique minimizer of $H(X_A \mid \tilde{Z}_V^{\tilde{S}_V})$ is any $S$ satisfying

$$H(X_A \mid \tilde{Z}_V^{\tilde{S}_V}) = H(X_A \mid \tilde{Z}_V), \tag{31}$$

which is equivalent to

$$I(\tilde{Z}_V^{\tilde{S}_V}; X_A) = I(\tilde{Z}_V; X_A). \tag{32}$$

Thus for each sample, $I(\tilde{Z}_V^{S_V(\tilde{z}_V, \tilde{z}_A)}; X_A) = I(\tilde{Z}_V; X_A)$, completing the proof. $\qquad\square$

The following lemma shows that adding an $\ell_1$-penalty on the mask encourages sparsity: the optimal mask discards all non-shared coordinates and exactly recovers the minimal shared block of the variables produced by the neural network.

**Lemma 5** (Sparsity-Induced Minimality). *Fix any invertible $q_V$. Under Assumptions 3–5, the joint minimizer*

$$(M_V^*, g_A^*) = \arg \min_{M_V, g_A} \left\{ \mathbb{E}\big[-\log Q_{g_A}(X_A \mid M_V \odot \tilde{Z}_V)\big] \;+\; \lambda\, \mathbb{E}[\|M_V\|_1] \right\} \tag{33}$$

*satisfies, for almost every sample,*

$$S_V^*(\tilde{Z}_V, \tilde{Z}_A) = \tilde{S}_V^\dagger, \quad I\big(\tilde{Z}_{V,j}; X_A \mid \tilde{Z}_V^{\tilde{S}_V^\dagger}\big) = 0 \quad \forall j \notin \tilde{S}_V^\dagger. \tag{34}$$

*That is, the mask prunes away all non-shared coordinates, recovering exactly the minimal shared block.*

*Proof of Lemma 5 (Sparsity-Induced Minimality).* Fix $q_V$. As before, by Decoder Universality (Lemma 3) the joint minimization over $(M_V, g_A)$ is equivalent to

$$\min_{M_V} \mathbb{E}\Big[ H\big(X_A \mid \tilde{Z}_V^{\tilde{S}_V}\big) + \lambda\, |\tilde{S}_V| \Big]. \tag{35}$$

Since $M_V$ can choose $\tilde{S}_V$ per sample (Assumption 4), we solve for each $(\tilde{z}_V, \tilde{z}_A)$:

$$\min_{\tilde{S} \subseteq [d]} f(\tilde{S}) \quad \text{where} \quad f(\tilde{S}) = H\big(X_A \mid \tilde{z}_V^{\tilde{S}_V}\big) + \lambda\, |\tilde{S}|. \tag{36}$$

For any $j \notin \tilde{S}$, adding $j$ changes $f$ by

$$f(\tilde{S} \cup \{j\}) - f(\tilde{S}) = -I\big(\tilde{Z}_{V,j}; X_A \mid \tilde{Z}_V^{\tilde{S}_V}\big) + \lambda. \tag{37}$$

By Assumption 5, $I\big(\tilde{Z}_{V,j}; X_A \mid \tilde{Z}_V^{\tilde{S}_V}\big) \leq \Delta_{V,j}([d]) < \lambda$, so $f(S \cup \{j\}) > f(S)$ and no non-blanket index is added. Similarly, for any $i \in \tilde{S}$, dropping $i$ changes $f$ by

$$f(\tilde{S} \setminus \{i\}) - f(\tilde{S}) = I\big(\tilde{Z}_{V,i}; X_A \mid \tilde{Z}_V^{\tilde{S} \setminus \{i\}}\big) - \lambda, \tag{38}$$

and Assumption 5 ensures this is positive for all $i \in \tilde{S}_V^\dagger$. Hence the unique minimizer is $\tilde{S} = \tilde{S}_V^\dagger$, and $I\big(\tilde{Z}_{V,j}; X_A \mid \tilde{Z}_V^{\tilde{S}_V^\dagger}\big) = 0$ for $j \notin \tilde{S}_V^\dagger$, as required. $\qquad\square$

The following theorem shows that, when jointly optimizing encoders, masks, and decoders with our bidirectional objective over both video and audio representations, the global minimizer precisely aligns and recovers the shared latent blocks—i.e. it achieves exactly the block-alignment specified in Definition 2.

**Theorem 2** (Global Block-Alignment and Recovery). *Under Assumptions 1, 2, 3, 4 and 5, the global minimizer of Objective 7 yields $\left(\tilde{Z}_V^{\tilde{S}_V^*}, \tilde{Z}_A^{\tilde{S}_A^*}\right)$ that satisfies Definition 2.*

*Proof.* We decompose the total training objective into two symmetric parts,

$$\mathcal{L} = \mathcal{L}_{V \to A}(q_V, M_V, g_A) + \mathcal{L}_{A \to V}(q_A, M_A, g_V),$$

where, for instance,

$$\mathcal{L}_{V \to A}(q_V, M_V, g_A) = \mathbb{E}\Big[-\log Q_{g_A}\big(X_A \mid M_V(\tilde{Z}_V, \tilde{Z}_A) \odot q_V(Z_V)\big)\Big] + \lambda\,\mathbb{E}\big[\|M_V\|_1\big].$$

**1. Existence of an optimal block-aligned configuration.** By Assumption 2, there exist $(q_V^\dagger, M_V^\dagger)$ and $g_A^\dagger$ such that

$$M_V^\dagger(\tilde{z}_V, \tilde{z}_A) \equiv \tilde{S}_V^\dagger, \quad \mathcal{L}_{V \to A}(q_V^\dagger, M_V^\dagger, g_A^\dagger) = H\big(X_A \mid \tilde{Z}_V^{\tilde{S}_V^\dagger}\big) + \lambda\,|\tilde{S}_V^\dagger|.$$

The same argument applies to the audio-to-video term, yielding $(q_A^\dagger, M_A^\dagger, g_V^\dagger)$.

**2. Optimality of the shared supports.** Fix any candidate $(q_V, M_V, g_A)$. First, for a fixed encoder $q_V$, Lemma 3 (Decoder Universality) shows that

$$\min_{g_A} \mathcal{L}_{V \to A}(q_V, M_V, g_A) = \mathbb{E}\big[H(X_A \mid \tilde{Z}_V^{\tilde{S}_V})\big] + \lambda\,\mathbb{E}\big[|\tilde{S}_V|\big].$$

Lemma 4 then implies any minimizer $M_V$ must satisfy

$$I\big(\tilde{Z}_V^{\tilde{S}_V}; X_A\big) = I\big(\tilde{Z}_V; X_A\big).$$

Next, we allow $q_V$ itself to vary. By Assumption 2, the encoder family contains some $q_V^\dagger$ that minimizes the conditional entropy $\mathbb{E}[H(X_A \mid \tilde{Z}_V^{\tilde{S}_V^\dagger})]$. Lemma 5 ensures the unique sparsest choice of $\tilde{S}_V$ is $\tilde{S}_V^\dagger$. Altogether, when $\lambda$ is sufficiently small, the global minimizer of the video→audio term satisfy

$$q_V^* = q_V^\dagger, \quad M_V^*(\cdot) = \tilde{S}_V^\dagger.$$

An identical chain of reasoning on $\mathcal{L}_{A \to V}$ yields

$$q_A^* = q_A^\dagger, \quad M_A^*(\cdot) = \tilde{S}_A^\dagger.$$

Therefore, at the global optimum both pairs $\left(\tilde{Z}_V^{\tilde{S}_V^*}, \tilde{Z}_A^{\tilde{S}_A^*}\right)$ thus coincide with the unique minimal sufficient latents $\left(\tilde{Z}_V^{\tilde{S}_V^\dagger}, \tilde{Z}_A^{\tilde{S}_A^\dagger}\right)$, so they satisfy Definition 1. $\qquad\square$

# C   Cascade Diffusion Model

Building on the aligned latent representations from Stage I, we generate video and audio in a cascaded manner using independently finetuned single-modal diffusion models. As illustrated in Figure 3 (II), the process begins by generating a video from a text prompt using a video latent diffusion model. The resulting visual latent $z_v^T$ is then adapted through a lightweight projection network $\mathcal{P}_\theta$, producing an audio-guiding latent $\tilde{z}_a$ that encodes visually grounded cues. This latent, along with the original text embedding, conditions the subsequent audio generation. By structuring the process in this cascaded fashion, we ensure that the audio is temporally and semantically aligned with the generated video content. Notably, the pretrained diffusion backbones remain frozen during training, and only the adapters are updated, preserving modularity and enabling efficient adaptation to downstream multimodal generation tasks.

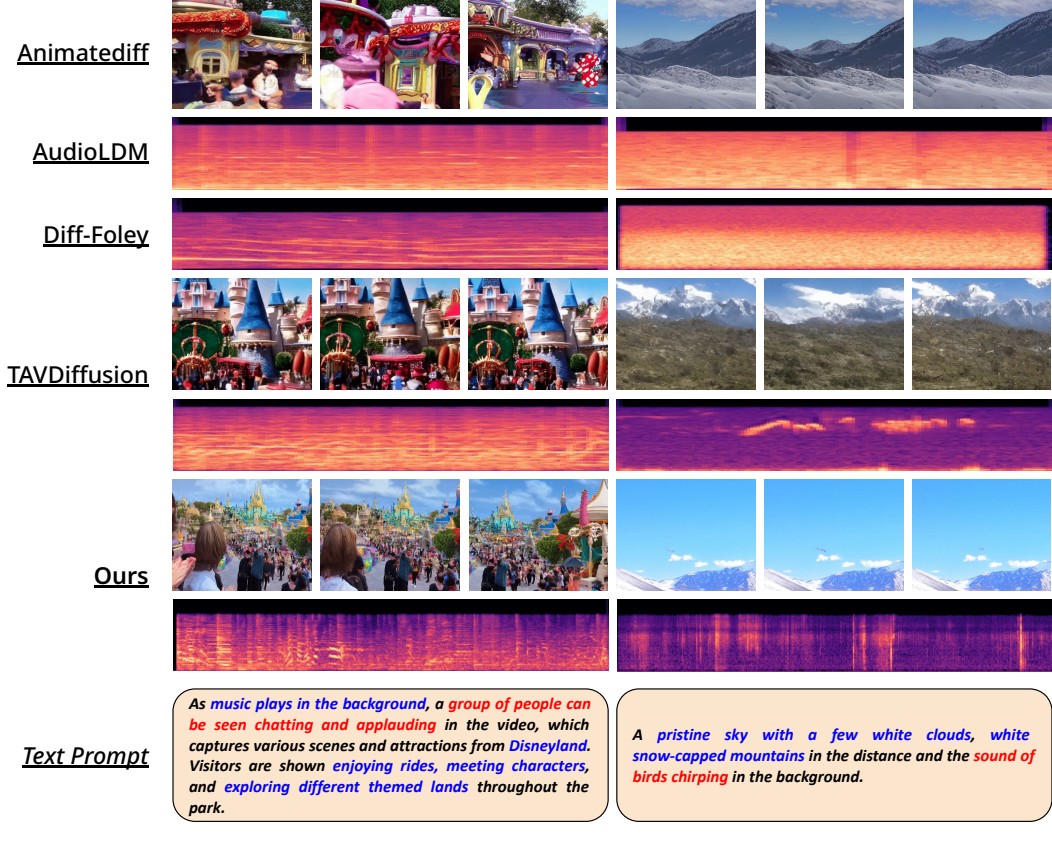

Figure 7: Additional Text-to-Audio-Video generation results compared with other baselines. We use the same text prompt as in [33] for our demonstration and compare the method against multiple baselines (Animatediff [8], AudioLDM [26], Diff-Foley [31], and TAVDiffusion [33]).

**Diffusion Formulation** Let $x_t$ denote the input text prompt, which is encoded via a pretrained text encoder $f_t(\cdot)$ to obtain $z_t = f_t(x_t)$. The video generation begins by sampling Gaussian noise $z_v^0 \sim \mathcal{N}(0, I)$, which is progressively denoised through the reverse diffusion process:

$$z_v^{t-1} = \frac{1}{\sqrt{\alpha_t}} \left( z_v^t - \frac{1 - \alpha_t}{\sqrt{1 - \bar{\alpha}_t}} \cdot \epsilon_{\theta_v}(z_v^t, z_t, t) \right) + \sigma_t \cdot \epsilon, \quad \epsilon \sim \mathcal{N}(0, I). \tag{39}$$

Once the final video latent $z_v^T$ is obtained, it is projected into an audio-guiding latent $\tilde{z}_a = \mathcal{P}_\theta(z_v^T)$. Audio generation is then conditioned on both $\tilde{z}_a$ and $z_t$ using an analogous denoising process:

$$z_a^{t-1} = \frac{1}{\sqrt{\alpha_t}} \left( z_a^t - \frac{1 - \alpha_t}{\sqrt{1 - \bar{\alpha}_t}} \cdot \epsilon_{\theta_a}(z_a^t, \tilde{z}_a, z_t, t) \right) + \sigma_t \cdot \epsilon, \quad \epsilon \sim \mathcal{N}(0, I). \tag{40}$$

The generated latents are subsequently decoded using pretrained decoders to obtain the final outputs: $\hat{x}_v = g_v(z_v^T)$ and $\hat{x}_a = g_a(z_a^T)$.

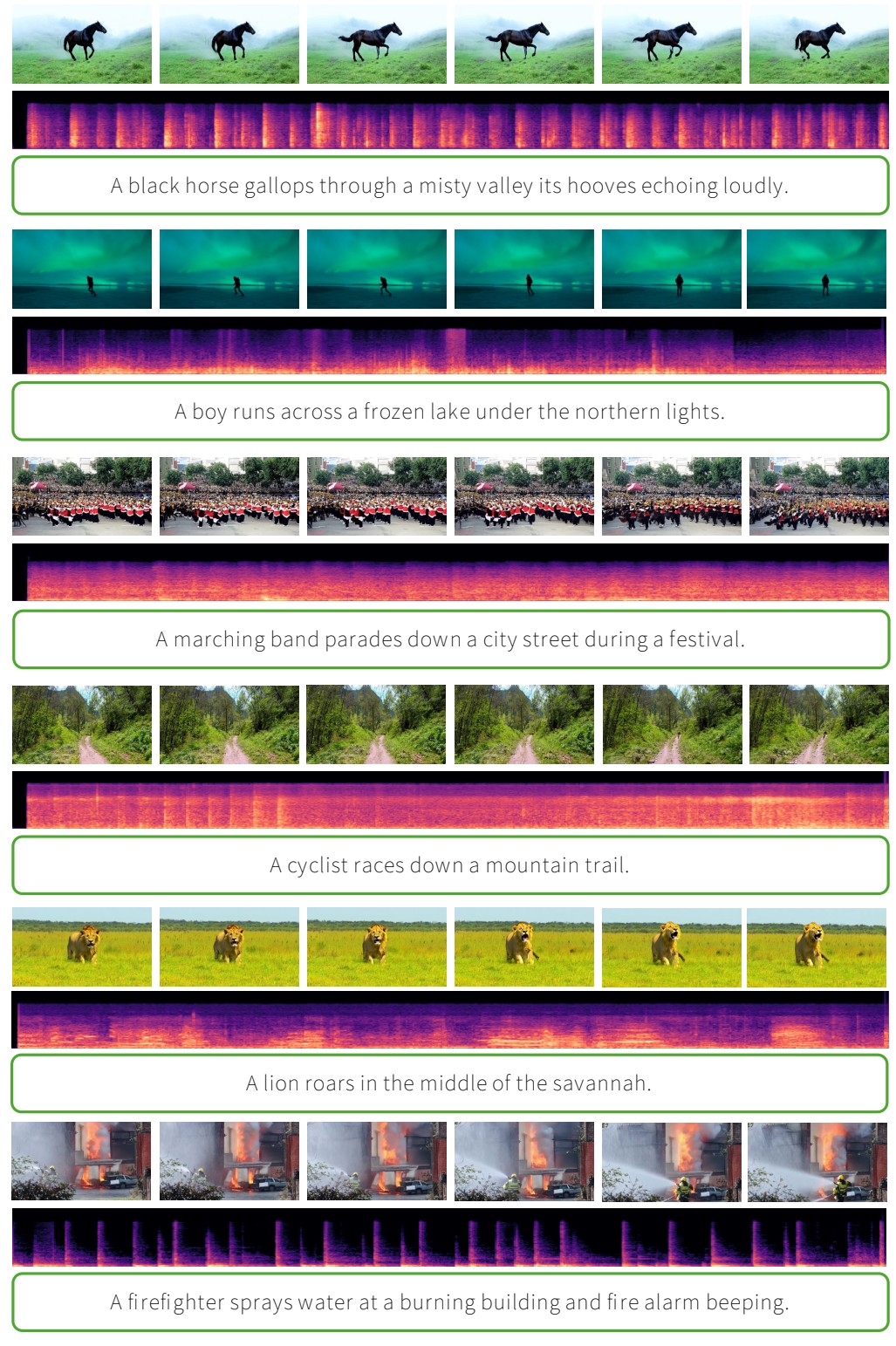

Figure 8: Additional Text-to-Audio-Video generation results by our proposed framework.

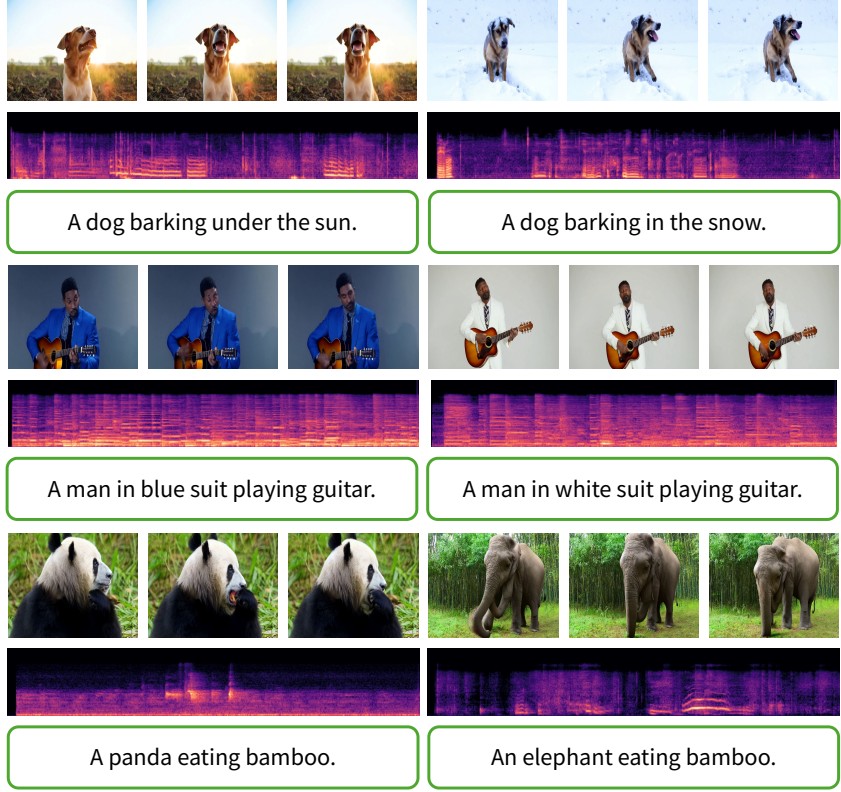

Figure 9: Change of audible or non-audible attributes to the generative results.

## D  Additional Results

We present additional visualizations of our T2AV-generated results in Figure 7. Using the same text prompts as those in [33], we generate audio-video pairs and compare them against existing baselines. In the *Disneyland* scene, our model produces vertically structured spectrogram features that plausibly correspond to discrete auditory events such as applause or exclamations, reflecting a diverse and dynamic soundscape. In the *sky with bird chirping* scene, we observe consistent, rhythmic patterns in the spectrogram that align with natural bird calls, indicating accurate temporal grounding of audio events. These results highlight the model's ability to synthesize semantically coherent and temporally aligned audio conditioned on visual content. Additional examples are provided in Figure 7.

## E  Sensitivity Test

We conduct a sensitivity analysis to probe how the system responds to prompt variations that (i) alter non-audible attributes (e.g., background, static non-audible objects) while keeping the sound-causing event unchanged, and (ii) alter audible attributes (e.g., object/action that produces sound) while holding non-audible details fixed. As shown in Figure 9, for each prompt pair, we generate matched video-audio samples under identical random seeds. Qualitatively, we visualize spectrograms alongside key video frames to inspect whether non-audible edits leave the soundtrack invariant and whether audible edits induce commensurate changes in rhythm and energy. Ideally, non-audible edits should produce minimal shifts in audio metrics, while audible edits should yield significant, directionally consistent changes. This analysis clarifies which aspects of the prompt our model treats as causally relevant for sound synthesis and highlights residual entanglements where visual-only edits still perturb the audio.

| $\alpha$ | AVH↑ | CAVP↑ | FAD↓ |
|---|---|---|---|
| 0.001 | 0.174 | 0.150 | 5.52 |
| 0.01 | 0.197 | 0.159 | 5.60 |
| 0.1 | 0.206 | 0.165 | 5.49 |
| 1 | 0.199 | 0.154 | 5.31 |

Table 4: Grid search summary for $\alpha$.

| $\lambda$ | AVH↑ | CAVP↑ | FAD↓ |
|---|---|---|---|
| 1 | 0.208 | 0.161 | 7.03 |
| 5 | 0.206 | 0.165 | 5.49 |
| 10 | 0.174 | 0.150 | 7.42 |
| 100 | 0.172 | 0.141 | 6.97 |

Table 5: Grid search summary for $\lambda$.

## F    Hyperparameter Studies

We study how the alignment weight $\alpha$ and sparsity weight $\lambda$ affect performance in Table 4 and 5. We sweep $\alpha \in \{0.001, 0.01, 0.1, 1\}$ and $\lambda \in \{1, 5, 10, 100\}$ on a validation split, measuring AVHScore and CAVP similarity (alignment), FAD (audio quality). In unidirectional setting the FVD is not affected by alignment. We Find that: Reducing $\alpha$ too much weakens the latent coupling, leading to semantic drift between modalities (e.g., misaligned motion and sound). Excessively high $\lambda$ over-prunes the mask. Disabling sparsity results in overly dense masks, which fail to isolate cross-modal signals and slightly reduce performance in AVHScore and CAVP similarity.

## G    Limitations

While our cascaded T2AV framework demonstrates strong performance in both generative quality and cross-modal alignment, several limitations remain. First, the reliance on sequential generation, where video is produced before audio, introduces unidirectional dependency that may constrain expressiveness in audio-visual co-synchronization, particularly for content requiring tight mutual feedback. Second, the alignment mechanism is trained on temporally segmented clips, which may limit its ability to generalize to complex or highly dynamic temporal structures in longer sequences. Third, although we finetune the diffusion models to better adapt to generated embeddings, the overall generation quality remains bounded by the capacity and resolution of the pretrained backbones, which are not jointly optimized with the alignment modules. Finally, our framework assumes the availability of high-quality paired data for training, extending it to settings with weak or noisy supervision remains an open challenge.

## H    Social Impact and Safeguards

Our work focuses on improving the quality and coherence of text-to-audio-video (T2AV) generation through aligned latent representations and cascaded diffusion models. While such generative capabilities hold potential for positive applications in assistive technologies, content creation, and education, they also raise ethical concerns related to misinformation, deepfakes, and unauthorized content synthesis. In particular, the ability to generate realistic audio-visual content conditioned on arbitrary text inputs could be misused to fabricate misleading media or impersonate individuals. To mitigate these risks, we recommend deploying the model with usage constraints such as content watermarking, access control mechanisms, and human-in-the-loop verification. Moreover, our system is not trained on or intended for real-person likeness reproduction, and safeguards should be established before applying it in socially sensitive domains. We also advocate for transparency in model provenance and responsible dataset curation, ensuring compliance with copyright, privacy, and fairness standards.

