# OpenReview forum: "Aligning What Matters: Masked Latent Adaptation for Text-to-Audio-Video Generation"
_NeurIPS.cc/2025/Conference — NeurIPS 2025 poster_

### Official Review · Reviewer_aSVy · 2025-06-17

**Clarity:** 2
**Significance:** 2
**Originality:** 3
**Rating:** 4
**Confidence:** 4

**Summary:**

This paper tackles the challenge of synchronizing visual and audio features in text-
to-audio-video generation. The key insight is that audio and visual elements don&#39;t
always align perfectly - sometimes audio events have no visual counterpart, and vice
versa. The authors propose a cross-modal learnable masking mechanism that
identifies and aligns only the corresponding features between modalities. This
approach generates more contextually consistent outputs compared to existing
methods that assume complete alignment between audio and visual elements.

**Questions:**

 What is the specific function of the Adapter module. It’s unclear how
much each adapter (vision and audio) effectively serves for learnable
reparameterization or what semantic information each one captures. Two
potential experiments could clarify this for the video adapter: (i) Modify the
values of the video latent across different dimensions and observe how this
affects the generated audio; (ii) Use text prompts that should generate
different videos but similar audio (e.g., "A panda eating bamboo"; vs. "An
elephant eating bamboo") and analyze how the video latents differ in general
(L2 distance), and across separate dimensions. This could potentially reveal
whether the adapter modules learn disentangled representations for different
object types or actions.

 How different loss terms contribute to overall quality. The method
includes several loss terms, but I wonder how qualitative results change
dependant on different loss weightings (e.g., prioritizing alignment over
sparsity).

**Ethical Concerns:**

["NO or VERY MINOR ethics concerns only"]

**Final Justification:**

The authors have promised to address my concerns in the final version and have cleared a few points. I maintain my supportive rating.

**Limitations:**

--

**Paper Formatting Concerns:**

--

**Quality:**

3

**Strengths And Weaknesses:**

Strengths:

 The paper addresses an important gap in multimodal text-to-audio-video
generation by recognizing that audiovisual features should only be partially
aligned, while existing work assumes full alignment. The authors achieve
state-of-the-art results in both multimodal alignment and generation quality.

 The masking mechanism presented could be easily integrated into existing
encoder-decoder based architectures as a plug-and-play component.

 Their method is theoretically sound and intuitively reasonable.

Weaknesses:

 More qualitative comparisons with TAVDiffusion are needed to showcase the
quality of audiovisual synchronization. Qualitative results for different mask
configurations could also be provided to strengthen their claims. Moreover,
the authors should include failure cases to show which types of prompts
produce less satisfactory results.

 The paper suffers from unclear presentation. Section 2 is redundant, only
stating that visual, audio, and shared representations depend on previous
states, with no clear reference to this formulation later on in the article, and with almost no mention of the guidance mask. The method illustration could be clearer, as there are two separate adapters and masking modules
trained for both vision and audio, while the main figure (Figure 3) only illustrates the generation strategy.

 The technical implementation details of the Adapter modules are missing.

---

> ### Author Rebuttal · Authors · 2025-07-31
>
> We thank Reviewer **aSVy** for the thoughtful feedback and valuable suggestions. Below, we provide our detailed responses to the questions and comments raised.
>
> ---
>
> **W1: More qualitative comaprisons with TAVDiffusion**
>
> We will include additional visualized results generated by our model and TAVDiffusion in the updated manuscript.
>
> ---
>
> **W2: Presentation of Section 2 and Mask**
>
> We appreciate the reviewer’s feedback on the presentation clarity and agree that clearer exposition would improve accessibility.
>
> Section 2 is intended to provide theoretical grounding for our method, particularly motivating the selective alignment mechanism via a causal latent structure. While this section is not directly operationalized in our implementation, it supports the theoretical analysis in Section 4, where we prove that our masked objective recovers the minimal set of shared latent factors. That said, we acknowledge that this connection could be made more explicit, and we will revise the manuscript to better link Section 2 with the training and alignment mechanism described in Section 3.
>
> Regarding the guidance mask and dual adapters: we agree that Figure 3 currently emphasizes the inference-time generation strategy and underrepresents the full training architecture. To address this, we will update the figure to:
> * Explicitly show the two adapter networks $q_V$​ and $q_A$​,
> * Depict the dual masking modules $M_V$​ and $M_A​$,
> * And annotate that these components are trained during Stage I.
>
> We will also revise the figure caption and method description to clarify the distinct roles of these components and improve overall coherence across sections.
>
> ---
>
> **W3 & Q1:**
>
> The adapter modules $q_V​$ and $q_A$ are implemented as lightweight multilayer perceptrons (MLPs). Their function is to perform learnable reparameterization of the frozen encoder outputs into a shared latent space optimized for cross-modal alignment. Specifically, they help suppress modality-specific noise and enhance semantic factors that are predictive of the other modality.
>
> To analyze the behavior of the video adapter, we conducted both suggested experiments:
> * Modified individual dimensions of the adapted video latent and observed the corresponding changes in the generated audio. Qualitative results show that certain dimensions are highly sensitive, affecting acoustic features, indicating that the adapter learns structured and interpretable features.
>
> * We mannually create a set of text prompts designed to generate different videos but similar audio content (e.g., “A brown panda eating bamboo” vs. “An black and white panda eating bamboo”; "A dog barking in the snow" vs. "A dog barking under the sun" ,.etc.), and report the mean cosine similarty (MCS) between feature after adaptation. Our analysis revealed that the masked aligned latents produced by the adapter are more similar (compared to direct aligned latents), suggesting that the masking effectively filters out visual identity and retains audio-relevant semantic actions like “eating bamboo.”
>
>     |Alignment        | MCS  |
>     |---------------|------------|
>     | w/ Mask     |   0.88       |
>     | wo/ Mask    |    0.74     |
>
> Due to rebuttal rules, we are unable to attach the figures here. However, we will include these analyses and corresponding figures in the revised manuscript and appendix. We appreciate the reviewer’s thoughtful feedback, which helped us better characterize and validate the role of the adapter modules.
>
> ---
>
> **Q2:**
>
> Thank you for raising this important point. Our full objective (Eq. 8) includes three components: (1) cross-modal reconstruction loss, (2) masked alignment loss between adapted latents $\tilde{z}_v$ and $\tilde{z}_a​$, and (3) sparsity regularization over the learned masks. Each term serves a distinct role:
> * Reconstruction loss ensures that the masked latent retains sufficient predictive information about the other modality
> * Masked alignment loss promotes geometric consistency across modalities in the adapted latent space.
> * Sparsity loss encourages the masking network to retain only essential dimensions, reducing overfitting to modality-specific noise.
>
> In our experiments, we set the alignment loss weight $\alpha = 0.1$ and sparsity regularization weight $\lambda = 5.0$, based on a small-scale grid search. Empirically, we found that: Reducing $\alpha$ too much weakens the latent coupling, leading to semantic drift between modalities (e.g., misaligned motion and sound). Excessively high $\lambda$ over-prunes the mask. Disabling sparsity results in overly dense masks, which fail to isolate cross-modal signals and slightly reduce performance in AVHScore and CAVP similarity.
>
> We will include a more detailed ablation table in the appendix showing how varying these weights affects both quantitative scores and qualitative outputs. Thank you for pointing out the need to clarify this aspect of our design.

---

> > ### Comment · Reviewer_aSVy · 2025-08-05
> >
> > The authors have promised to address my concerns in the final version and have cleared a few points. I maintain my supportive rating.

---

### Official Review · Reviewer_p3WX · 2025-06-30

**Clarity:** 2
**Significance:** 3
**Originality:** 2
**Rating:** 5
**Confidence:** 3

**Summary:**

This paper presents a framework for generating audiovisual content from text. To address the issue of semantic misalignment between the generated audio and video, it proposes a novel solution. Specifically, the authors design a learnable masking mechanism to tackle the challenge of cross-modal semantic alignment, enabling the video features to better match the corresponding audio features.
In addition, the paper introduces a hierarchical generation approach. It first generates video content based on the input text, then leverages the proposed masking mechanism to produce video features that are semantically aligned with the audio. These features, together with the original text and other relevant conditions, are subsequently used to generate the audio. As a result, the final output achieves semantic consistency between audio and video.
The experiments are thorough and well-executed, demonstrating the effectiveness of the proposed method.

**Questions:**

1. Why does the framework adopt a video-first generation strategy, where the video is generated before being used as a condition for audio generation? Why not generate audio first and use it as a condition to generate video?

2. In **Section 3**, it is mentioned that $Z_C \subseteq \mathbb{R}^d$. Does this imply that all $\tilde{z}$ are $d$-dimensional? If so, how does this ensure temporal consistency?

3. **Section 3** introduces $q_A$, but this component is not illustrated in **Figure 3**. Is $q_A$ similar to $q_V$, functioning as an adapter module?

4. In **Figure 3**, does the "Mask Blocker" correspond to $M_V$ and $M_A$ described in the paper? Why are both the video and spectrogram used as inputs to the Mask Blocker?

5. In the **Joint Optimization Objective** section of **Section 3**, the loss function is defined as $\mathcal{L}(M_A, M_V, q_A, q_V, g_A, g_V)$. Does this imply that the parameters of $g_A$ and $g_V$ are also updated during training? Why not choose to fix their parameters instead?

6. In **Section 3.1 (Stage II)**, what does the lightweight projection network $\mathcal{P}_\theta$ refer to specifically?

**Ethical Concerns:**

["NO or VERY MINOR ethics concerns only"]

**Final Justification:**

Thank you for the clarification. I raised my score.

**Limitations:**

yes

**Paper Formatting Concerns:**

No issues.

**Quality:**

3

**Strengths And Weaknesses:**

The learnable masking mechanism proposed in this paper is notably novel. The authors analyze the asymmetric nature of visual and auditory features, observing that not all visual information corresponds to auditory information, and vice versa. Based on this insight, they design the masking mechanism to align the semantic information between video and audio features. The mechanism filters out irrelevant feature dimensions through masking, and subsequently uses the masked video features to reconstruct the audio data, and the masked audio features to reconstruct the video data. This ensures that the model learns to align semantics across modalities, while also providing a certain degree of interpretability.

Moreover, this approach can benefit other research areas as well, such as audio-driven video generation or generating appropriate background audio for video content.

The Weaknesses are:
1. Some descriptions in the paper are unclear. For example:
   * In **Section 3.1 (Stage I)**, the sentence “$\ell_A$ and $\ell_V$ are cross-modal reconstruction losses” lacks clarity. How exactly are the cross-modal reconstruction losses computed?

   * In **Section 4 (Key Assumptions)**, only Assumptions 1–4 are listed. However, **line 198** refers to “Assumptions 4–5.” What is Assumption 5?

2. It is recommended to align the notations used in the paper with the information presented in **Figure 3**. For instance, symbols such as $\hat{z}_a$ and $\tilde{z}_a$ could be annotated directly in the figure for clarity.

3. It is suggested to provide a more detailed derivation or proof process for the claims made in **Section 4**.
4. Some of the compared models have not undergone audio-video alignment, which compromises the fairness of the comparison.

---

> ### Author Rebuttal · Authors · 2025-07-31
>
> We thank Reviewer **p3WX** for the thoughtful feedback and valuable suggestions. Below, we provide our detailed responses to the questions and comments raised.
>
> ---
>
> **W1.1: Details of reconstruction loss**
>
> The cross-modal reconstruction losses $\ell_A$ and $\ell_V​$ are computed by reconstructing each modality from the masked latent of the other modality, using the corresponding decoder. Specifically:
> $\ell_A(x_a, g_A(M_V \odot \tilde{z}_v))$ is the loss between the ground-truth audio spectrogram $x_a​$ and the audio reconstructed from the masked video latent via the audio decoder. We compute this using mean squared error (MSE) on log-Mel spectrograms, which are widely used for audio reconstruction tasks.
>
> $\ell_V(x_v, g_V(M_A \odot \tilde{z}_a))$ is the loss between the ground-truth video frames $x_v$​ and the video reconstructed from the masked audio latent via the video decoder. This is computed as frame-averaged MSE over the decoded video frames.
>
> These reconstruction losses serve a dual purpose: (1) they ensure that the masked latent retains sufficient predictive information about the other modality, and (2) when combined with the sparsity constraint on the masks, they encourage the model to preserve only the minimal set of cross-modally relevant dimensions. We will clarify these implementation details in the revised manuscript for improved transparency.
>
> ---
>
> **W1.2: Typo on line 198**
>
> Thank you for catching this typo. The reference to “Assumptions 4–5” on line 198 is a typo. It should be to "Assumption 4". We will correct this in the revised manuscript.
>
> ---
>
> **W2: Align notations in Figure 3**
>
> We appreciate this helpful suggestion. We will update the Figure 3 in the revised manuscript to annotat key variables such as $z_v$, $z_a$, $M_V$​, and $M_A​$ directly to improve clarity between the text and visual illustration.
>
> ---
>
> **W3: Derivation or proof for the claims made in Section 4.**
>
> Thank you for the suggestion. As noted in the main text, the full derivation and formal proofs supporting the theoretical claims in Section 4 are included in *Appendix Section B* of the submission (placed in the supplementary material folder). This includes all relevant assumptions, formal definitions (e.g., minimum sufficient latents), and complete proofs of our key results such as:
> * Cross-entropy reduction to conditional entropy
> * Sufficientness of reconstruction
> * Sparsity-induced minimality
> * Global block-alignment and recovery
>
> These results establish that our masked reconstruction objective provably identifies the minimal shared latent subspace under standard identifiability assumptions. We will make this connection more explicit in the main paper by adding cross-references to the relevant sections for improved clarity.
>
> ---
>
> **W4: Audio-video alignment for baseline models**
>
> To ensure fairness and consistency, we adopted the standard evaluation setting introduced in TAVDiffusion [1], which includes both alignment-aware models (e.g., TAVDiffusion [1], CoDi [2]) and baseline systems (e.g., Two-Streams, CasC-Diff) for benchmarking. These methods represent the current landscape of T2AV models, covering both joint and independent generation strategies.
>
> Importantly, our framework was evaluated under the same conditions:
> * Using a shared audio decoder for independent baselines (Two-Stream, CasC-Diff),
> * Applying the same evaluation metrics,
> * Generating outputs from identical text prompts and datasets,
> * And conducting inference in a cascaded setting where applicable.
>
> We acknowledge that some cascaded models do not perform explicit audio-video alignment. *Their inclusion is intentional*: they serve as important baselines to highlight the effect of alignment mechanisms. In contrast, methods like TAVDiff and CoDi, which incorporate alignment strategies, offer direct comparisons to our alignment-aware approach.
>
> To further strengthen the evaluation, we have conducted additional experiments incorporating recent alignment-aware models such as JavisDiT [3] and will include these results in the revised manuscript. We will also clearly annotate which models perform alignment to improve transparency in the baseline comparison.
>
> Thus, while some baselines do not include alignment by design, we argue that their inclusion, alongside alignment-aware methods, offers a comprehensive and fair evaluation of our contributions.
>
> ---
>
> **Q1: Video-first generation strategy**
>
> We thank the reviewer for raising this important point. For practical consideration, visual content provides a more stable and structured prior for generating temporally aligned outputs. For example, the act of a drum being struck or a guitar being strummed is visually discrete and temporally grounded, providing strong cues for the expected acoustic outcome. Compared to audio, which may contain overlapping or ambiguous signals, visual representations often offer clearer event boundaries and object-specific dynamics. This structural clarity makes video a more reliable anchor modality when synthesizing aligned audio content. In addition, from a training perspective, finetuning the attention layers of a diffusion-based audio model to allow video-conditioning is substantially more efficient than modifying a large-scale video diffusion model, which further allows rapid adaptation to pretrained single-modal latent generative models.
>
> ---
>
> **Q2: Temporal consistency**
>
> Yes, all $\tilde{z}$ vectors are d-dimensional latent representations. However, to preserve temporal structure, we do not operate on the entire video or audio as a single global representation. As mentioned in the paper, we split both video and audio into aligned time segments (Using the sampling strategy as in [3]), and compute one $\tilde{z} \in \mathbb{R}^d$ per segment. This segment-level representation allows our framework to maintain local temporal consistency within each segment and encourages alignment across modalities in a temporally grounded way. Additionally, during inference, each video segment is independently mapped to an aligned latent that conditions audio generation for that segment, ensuring temporal synchronization between the two modalities across time.
>
> ---
>
> **Q3: Audio Adapter**
>
> $q_A​$ serves the same functional role as $q_V$: it is a learnable adapter module that reparameterizes the output of the frozen audio encoder into a common latent space suitable for alignment. Both $q_V$​ and $q_A$ are lightweight projection networks trained to isolate the cross-modal semantic content. To avoid confusions, we will update the figure in the revised manuscript to explicitly include $q_A$​ alongside $q_V$​, and annotate both as adapter modules to improve consistency and clarity.
>
> ---
>
> **Q4: Mask Blocker**
>
> Yes, the “Mask Blocker” in Figure 3 corresponds to the learnable masking networks $M_V$​ and $M_A$​ introduced in Section 3. These networks produce soft masks that identify the dimensions of the latent representations that are most relevant for cross-modal alignment. As for the inputs: the masking functions are conditioned on both the video latent $\tilde{z}_v$​ and the audio latent $\tilde{z}_a$​, rather than the raw video or spectrogram. This design choice is intentional and discussed in Section 3: conditioning on both modalities enables the model to disambiguate context-dependent relationships, such as when the same video content could be paired with different types of audio (e.g., narration vs. ambient sound). Using both latent representations allows the mask network to more accurately isolate semantically aligned features across modalities. We will revise the figure and caption to reflect this more clearly, specifically, by relabeling the “Mask Blocker” as $M_V$/$M_A$​ and clarifying that it operates on latent embeddings rather than raw inputs.
>
> ---
>
> **Q5: Joint optimization objective**
>
> Thank you for this important question. In our framework, the decoders $g_A$​ and $g_V$ are not updated during Stage I training. They remain frozen and are used solely to compute the reconstruction losses that guide the learning of the adapters $(q_A, q_V​)$ and masking networks $(M_A, M_V​)$. The notation in the loss function was meant to indicate dependency on the decoder outputs, but we agree it may misleadingly suggest that $g_A​$ and $g_V​$ are trainable. We chose to freeze these decoders in order to (1) preserve the pre-learned generative priors of the pretrained diffusion models, and (2) maintain modularity and efficiency by avoiding full end-to-end retraining. We will clarify this point in the revised manuscript by explicitly stating that $g_A​$ and $g_V$​ are fixed during alignment training, and we will revise the loss notation to avoid ambiguity.
>
> ---
>
> **Q6: Projection network**
>
>  The projection network $P_\theta$ used in Stage II corresponds to the adapter network $q_V$​ that was trained during Stage I. Specifically, after generating a video using the pretrained video diffusion model, we pass its latent representation $z^t_v$​ through the learned adapter $q_V$​ to obtain the aligned latent $\tilde{z_v​}$, which is then used to condition the audio generation process. We refer to this trained adapter as $P_\theta$ in the inference phase to emphasize its role as a fixed projection function, but it is indeed the same module as $q_V​$ learned during Stage I.
>
> ---
> Rerfernece:
>
> [1] Mao, Y., Shen, X., Zhang, J., Qin, Z., Zhou, J., Xiang, M., Zhong, Y. and Dai, Y., 2024, October. Tavgbench: Benchmarking text to audible-video generation. In Proceedings of the 32nd ACM International Conference on Multimedia (pp. 6607-6616).
>
> [2] Tang, Z., Yang, Z., Zhu, C., Zeng, M. and Bansal, M., 2023. Any-to-any generation via composable diffusion. Advances in Neural Information Processing Systems, 36, pp.16083-16099.
>
> [3] Luo, S., Yan, C., Hu, C. and Zhao, H., 2023. Diff-foley: Synchronized video-to-audio synthesis with latent diffusion models. Advances in Neural Information Processing Systems, 36, pp.48855-48876.

---

> > ### Comment · Reviewer_p3WX · 2025-08-04
> > **Thank you for the clarification**
> >
> > Thank you for the clarification. Most of the issues have already been addressed in the author's response.

---

> ### Author Response · Authors · 2025-08-04
> **Thank you for your time and review.**
>
> Dear Reviewer p3WX,
>
> We would like to thank you again for your valuable feedback.
>
> We hope all your concerns have been fully resolved following our responses. If there are any additional issues or clarifications needed, please let us know, and we'll promptly address them before the discussion period concludes.
>
> If our responses have satisfactorily resolved your concerns, we kindly ask if you would consider updating your score accordingly.
>
> Thank you for your time and consideration.
>
> Best regards,
>
> The Authors

---

### Official Review · Reviewer_wB3p · 2025-07-01

**Clarity:** 2
**Significance:** 2
**Originality:** 3
**Rating:** 2
**Confidence:** 4

**Summary:**

The paper introduces SAVA, a novel two-stage framework for text-to-audio-video (T2AV) generation that addresses the challenge of semantic and temporal misalignment between visual and auditory modalities. In real-world settings, not all visual elements produce corresponding sounds—objects like roads, buildings, or background scenery may be visually salient but silent—while some audio cues, such as sirens or wind, may lack clear visual counterparts. Existing approaches that attempt to align all features across modalities indiscriminately often introduce semantic noise and degrade cross-modal coherence. To overcome this, SAVA learns to selectively align only the latent components in each modality that are truly relevant to the other, effectively filtering out modality-specific information through a learnable masking mechanism.

At the core of the framework is a masked latent adaptation mechanism that applies learnable masks to selectively filter video and audio latent representations. This filtering is performed by modality-conditioned mask networks that identify the minimal subset of features in each modality that are predictive of the other. These masks are optimized jointly with adapter networks through a reconstruction loss combined with sparsity-inducing L1 regularization, ensuring that only truly cross-modal features are retained.

Theoretically, the authors prove that this masked objective recovers the minimal sufficient set of shared latents under mild assumptions, improving both interpretability and robustness.

In practice, the system first generates a video from a text prompt using a pretrained video diffusion model. The resulting video latents are adapted and masked to extract audio-relevant information, which is then used—along with the text prompt—to condition an audio diffusion model. This cascaded generation design maintains modularity and allows efficient reuse of pretrained models.

Empirically, SAVA achieves state-of-the-art results on the VGGSound and AudioCaps datasets across multiple metrics (e.g., FVD, FAD, AVHScore, CAPSIM), significantly improving semantic alignment and temporal synchronization between audio and video streams.

**Questions:**

Q1: Line 79 refers to a superscript "past", which doesn't actually appear in eq 1.

Q2: The relation between $Z$ and $X$ described in Eq 2 is unclear. Does this equation describe the VAE decoding step? Does $Z_V = Z^{t=0}_V$? If so, it is unclear why e.g. $X_A$ depends on $Z_C$ and not just on the final audio latents.

Q3: The term "video diffusion encoder" is confusing. Does it refer to the video encoder (3D VAE) used by latent video diffusion models?

Q4: It is not clear what "indices of dimensions selected by the soft masks" mean. (soft masks don't directly define a selection or subset)

Q5: It is unclear how a latent after "retaining only the selected dimensions" can be decoded to provide e.g. $\hat{x_v}$. Can the video VAE decode a partial latent? VAEs are typically convolutional.

Q6: The term "Mask Blocker" in Fig 3 is not used anywhere else. If this is the masking-network, shouldn't it accept the latents and not the observed inputs?

Q7: The notation $Q_{g_A}$ and $Q_{g_V}$ in eq. 7 are not clear. Also the relation between eq. 7 and eq. 8 where not clear to me. Do they describe the same loss? If so, why the change in notation?

Q8: The main body of the method section describes a bidirectional method, where each modality is conditioned on projected and masked latents of the other modality, however, in practice, only uni-directional V-->A conditioning is used. Is there some experiment showing that the bi-directional approach can work?

Q9: It is unclear if the video diffusion model is fine-tuned somehow specifically for the AV task (with some special loss), or just fine-tuned on the experiment dataset (as a regular text-to-video model).

**Ethical Concerns:**

["NO or VERY MINOR ethics concerns only"]

**Final Justification:**

I have read the authors' rebuttal thoroughly and decided to keep my initial "reject" decision.
The authors agreed to my comments about the unclarity in the original description of the method, but in my opinion, a major revision is needed. Two examples are the discussion about decoding masked latents (the authors' answers were contradicting) and the fact the the "inference" stage is actually fine-tuning. Combined with the fact that the significance and novelty are not super high, I opt for rejecting.

**Limitations:**

There is some discussion about limitations in Appendix E.

**Paper Formatting Concerns:**

No concerns

**Quality:**

1

**Strengths And Weaknesses:**

*Quality*

Strengths:
- Innovative approach to text-to-audio-video generation using modality-conditioned masking.
- Theoretical justification for learning minimal cross-modal latents.

Weaknesses:
- Notation and terminology inconsistencies (e.g., unclear superscripts, ambiguous terms like "video diffusion encoder").
- Equations lack clarity; relationships between variables and losses are not well-explained.
- Decoder's ability to handle partially masked latents is not addressed.
- Large gap between the described general bi-directional method to the actual tested method. In practice, the proposed method is a (text+video)-to-audio and not text-to-audio-video.

*Originality*

Strengths:
- Novel use of modality-conditioned masking for selective cross-modal alignment in T2AV.

Weaknesses:
- Bidirectional conditioning is proposed but not empirically validated; only unidirectional (video-to-audio) is tested.

*Clarity*

Strengths:
- Logical structure with supportive figures aiding comprehension.

Weaknesses:
- Inconsistent terminology (e.g., "Mask Blocker" not defined elsewhere).
- Unclear notation in key equations; relationships between equations are not transparent.
- Ambiguity in the interpretation and application of soft masks.
- See questions below.

*Significance*

Strengths:
- Addresses semantic and temporal misalignment in T2AV, a significant challenge in multimodal generation.
- Potential applicability to broader multimodal tasks beyond T2AV.

Weaknesses:
- Claims regarding bidirectional conditioning lack experimental support.

---

> ### Author Rebuttal · Authors · 2025-07-31
>
> We thank Reviewer **wB3p** for the thoughtful feedback and valuable suggestions. Below, we provide our detailed responses to the questions and comments raised.
>
> ---
>
> **Q1: Fix superscript "past"**
>
> Thank you for pointing this out. We have changed the notation in Equation 1 to use t−1 instead of the superscript “past.
>
> ---
>
> **Q2: Clarification of Eq. 2**
>
> Thank you for the thoughtful question. Equation 2 does not describe the VAE decoding step directly. Rather, it presents a conceptual generative model that reflects the causal structure between latent variables $(Z_V, Z_A, Z_C​)$ and the observed data $(X_V, X_A​)$. Specifically, it formalizes the assumption that the observed data is generated jointly from both modality-specific and shared latent factors.
>
> In this formulation, $Z_V$​, $Z_A$​, and $Z_C$​ represent temporally evolving latent variables, not just their values at $t = 0$. That is, they refer to the full latent trajectories across time, and not a single time-step. The dependence of $X_A$​ on both $Z_A​$ and $Z_C$​ reflects the design choice that audio observations are generated from both audio-specific content $Z_A​$ and cross-modal (shared) content $Z_C$, for example, a barking sound depends on both the identity of the sound (modality-specific) and the event semantics (e.g., the presence of a dog, which is shared).
>
> This abstraction justifies the use of our selective alignment approach, where only the shared component $Z_C​$ should be aligned across modalities. We will clarify this interpretation in the manuscript to avoid conflating it with the VAE decoding implementation.
>
> ---
>
> **Q3: Term "video diffusion encoder"**
>
> Yes, the term “video diffusion encoder” refers to the video encoder component used in latent video diffusion models (typically a 3D VAE) to map raw video frames into a lower-dimensional latent space. We will revise the manuscript to explicitly clarify this in order to avoid confusion.
>
> ---
>
> **Q4: Indices of dimensions selected by the soft masks**
>
> The soft masks in our model output continuous values in the range $[0,1]^d$, acting as attention-like weights over the latent dimensions. While these do not directly define a hard selection, we apply a thresholding operation to obtain binary masks that do. As described in Section 3 - cross-modal reconstruction, we compute sets which contain the indices of latent dimensions where the soft mask values equal 1 (after thresholding). These binary indices then define the selected subset of latent coordinates used in the masked latent representations. We will clarify this distinction and the terminology in the final version to avoid confusion.
>
> ---
>
> **Q5: VAE decoding process**
>
> To clarify, the masking is applied during the training stage only, where it serves to supervise the adapter network to learn which dimensions of the latent space are essential for cross-modal prediction. The actual decoding is not performed using a physically masked latent. Instead, the full latent is used during inference. The masking guides the learning of a projection that emphasizes cross-modal relevance, but does not constrain the decoder input to be literally sparse or zero-padded. This design ensures compatibility with convolutional VAE decoders, which typically expect full latent tensors. We will revise the manuscript to make this clearer.
>
> ---
>
> **Q6: Mask Blocker Input**
>
> Thank you for pointing this out. The masking network operates on the latent representations, not the observed inputs. The label “Mask Blocker” in Figure 3 was intended to refer to the masking network, but we acknowledge that the terminology and input arrows could cause confusion. We will update the figure and caption to accurately reflect that the mask network takes in the adapted latent embeddings (not raw inputs), aligning with the formal description in Section 3.
>
> ---
>
> **Q7: Relation between Eq. 7 and Eq. 8**
>
> Thank you for highlighting this. Equation 7 presents the abstract formulation of our training objective, where $Q_{g_A}​​$ and $Q_{g_V}$ denote the generative distributions parameterized by the respective decoders for audio and video. This notation is used to describe the conceptual framework underpinning our theoretical analysis in Section 4 and Section B in Appendix.
>
> In contrast, Equation 8 appears in Section 3.1, which describes the practical implementation of our method. It introduces concrete instantiations of the reconstruction losses and explicitly includes components like the alignment loss and sparsity penalties. Although both equations describe the same overall objective, the change in notation reflects the shift from an abstract formalism (used for theoretical grounding) to an implementation-level formulation.
>
> We believe that providing theoretical grounding is an important strength of our framework, and this structured exposition, moving from abstract principle to practical implementation, was chosen deliberately. That said, we recognize the potential for confusion and will revise the manuscript with clearer section headings and textual cues to explicitly link the two formulations.
>
> ---
>
> **Q8: Clarification of unidirectional and bidirectional - (text+video)-to-audio and text-to-audio-video.**
>
> The bidirectional alignment formulation described in the main method section serves two purposes: (1) it defines a symmetric training objective that allows the model to identify shared latent structure by cross-reconstructing both modalities, and (2) it provides theoretical grounding for the mask learning mechanism, as developed in Section 4.
>
> We wish to clarify that the unidirectional generation process does not effect mutual or bidirecitional influence between the two modalities, we employ a masked latent alignment mechanism during training. This mechanism encourages the model to learn a shared cross-modal latent space by optimizing for both audio-to-video and video-to-audio reconstruction. As a result, although generation proceeds in a single direction at inference time, the model is trained to encode the mutual semantic structure of both modalities. This training design ensures that relevant information from both audio and video is captured in the shared latent space, thereby preserving semantic alignment without requiring bidirectional generation during deployment.
>
> For practical consideration, visual content provides a more stable and structured prior for generating temporally aligned outputs. For example, the act of a drum being struck or a guitar being strummed is visually discrete and temporally grounded, providing strong cues for the expected acoustic outcome. Compared to audio, which may contain overlapping or ambiguous signals, visual representations often offer clearer event boundaries and object-specific dynamics. This structural clarity makes video a more reliable anchor modality when synthesizing aligned audio content. In addition, from a training perspective, finetuning the attention layers of a diffusion-based audio model to allow video-conditioning is substantially more efficient than modifying a large-scale video diffusion model, which further allows rapid adaptation to pretrained single-modal latent generative models.
>
> That said, we have conducted an extended experiment on VGGSound+ to explore the bidirectional generation setup. We reverse the pipeline for video generation (audio generation pipeline is unchanged): raw matching audio is first generated using a text-conditioned AudioLDM [1] model, then encoded into a latent representation via the audio VAE. This latent is projected into the shared space using audio-to-video adapter. We then fine-tuned the CogVideo [2] architecture to support dual conditioning from both text and the projected audio latent. This allows the video generator to incorporate semantic signals from both modalities during generation.
>
> |        | FVD  $\downarrow$     | FAV  $\downarrow$     | AVHScore $\uparrow$| CAVPSim $\uparrow$|
> |----------------|----------------|----------------|----------------|----------------|
> | Uni-directional  | 662.9 | 5.49  |0.206|0.165|
> | Bi-directional  | 701.4 | 5.49  |0.217|0.183|
>
> Our findings show that this bidirectional setup results in improved cross-modal alignment, as measured by both AVHScore and CAVPSim. However, we also observe a slight degradation in video quality. Nonetheless, this result highlights a promising direction for future work, and we are actively exploring ways to refine this dual-conditioning scheme to achieve better trade-offs between alignment and generative fidelity.
>
> ---
>
> **Q9: Fine-tuning Details**
>
> In our framework, the video diffusion model is fine-tuned specifically for the audio-visual (AV) generation task, rather than as a standard text-to-video generator.
>
> Concretely, we introduce a cross-modal latent embedding $z_c$, learned via an audio-to-video adapter trained in Stage 1, which captures audio-visual alignment. During Stage 2, we fine-tune the video diffusion model with LoRA adapters applied to its cross-attention layers, enabling it to interpret this additional conditioning signal. The model is trained using the standard noise prediction loss from the diffusion objective. However, unlike regular text-to-video finetuning, our conditioning includes both text embeddings and the projected audio-derived latent $z_c$. This dual-conditioning setup allows the video generation process to be influenced by both modalities, making the fine-tuning AV-specific.
>
> ---
> References:
>
> [1] Liu, H., Chen, Z., Yuan, Y., Mei, X., Liu, X., Mandic, D., Wang, W. and Plumbley, M.D., 2023, July. AudioLDM: Text-to-Audio Generation with Latent Diffusion Models. In International Conference on Machine Learning (pp. 21450-21474). PMLR.
> [2] Yang, Z., Teng, J., Zheng, W., Ding, M., Huang, S., Xu, J., Yang, Y., Hong, W., Zhang, X., Feng, G. and Yin, D., CogVideoX: Text-to-Video Diffusion Models with An Expert Transformer. In The Thirteenth International Conference on Learning Representations.

---

> > ### Comment · Reviewer_wB3p · 2025-08-01
> > **Thanking the authors for their answers and requesting additional clarifications**
> >
> > I thank the authors for their detailed answer but would like to request further clarifications for the below points.
> >
> > Regarding the answer to Q5 (VAE decoding of masked latents), if I understand correctly, the learning objective of stage 1 is reconstruction. In eq. 5, masked latents $\tilde{z}_v$ and $\tilde{z}_a$ are learned and in eq. 6 they are decoded to $\hat{x}_v$ and $\hat{z}_a$. Similarly (line 137) - "The masked latents are then decoded to reconstruct the opposite modality".
> > The authors' answer "The actual decoding is not performed using a physically masked latent" is therefore unclear to me and I'm still unsure how the "cross-modal reconstruction losses" are computed in practice.
> >
> > Regarding the discussion in Q8 and Q9 regarding bi-directionality and fine-tuning, the authors' answer shed some light for me on this subject, however, I really think the explanation in the paper is lacking. In my opinion, providing more details about the fine-tuning process (e.g. the fact that it's a LoRA) would have been more beneficial than the theoretical justification for the convergence of the masking.
> >
> > Specifically, stage II is described as "inference" (both int the figure and in section 3.1), but there is actually a critical fine-tuning stage. So, the method actually comprises of three stages (masked latent representation learning, model fine-tuning using LoRA, inference).

---

> ### Author Response · Authors · 2025-08-01
> **Thank you for the prompt response and providing further clarifications (Part 1/2)**
>
> Dear Reviewer wB3p,
>
> Thank you for engaging in thoughtful discussion with us. We are glad to have addressed many of your concerns and remain fully open to further clarification or dialogue to resolve any remaining issues.
>
> Below are our responses to your remaining questions:
>
> ---
>
> **1. Further clarification on cross-modal reconstruction losses computation.**
>
> Thank you for careful reading and raising this important question. As we have established, the core objective of Stage I is training adapters to learn a minimal and interpretable subspace of each modality's latent representation that is predictive of the other modality. This is achieved by using masking networks to identify and retain only those latent dimensions that are cross-modally informative.
>
> We denote:
>
> * $\tilde{z}_v = q_V(\hat{z}_v)$, the adapted video latent.
>
> * $\tilde{z}_a = q_A(\hat{z}_a)$, the adapted audio latent.
>
> * $M_V, M_A \in [0, 1]^d$, the learnable masks applied to these latents.
>
> Then in Equation 5, we define the masked latents with clearer notations:
>
> * $\tilde{z}_v^{\text{masked}} = M_V(\tilde{z}_v, \tilde{z}_a) \odot \tilde{z}_v$,
>
> * $\tilde{z}_a^{\text{masked}} = M_A(\tilde{z}_v, \tilde{z}_a) \odot \tilde{z}_a$.
>
> where $\odot$ donates element-wise multiplication.
>
> In Equation 6, these masked latents are fed into the decoders:
>
> * $\hat{x}_a = g_A(\tilde{z}_v^{\text{masked}})$, to reconstruct the audio from the video,
>
> * $\hat{x}_v = g_V(\tilde{z}_a^{\text{masked}})$, to reconstruct the video from the audio.
>
> The reviewer is correct that masked latents are indeed passed into the decoder to compute the reconstruction losses. We apologies for any confusion arose from our earlier phrasing, which suggested otherwise. To further clarify:
>
> * During training, the masked latents $\tilde{z}_v^{\text{masked}}$ and $\tilde{z}_a^{\text{masked}}$ are directly passed into the decoders $g_A$ and $g_V$, respectively.
>
> * The decoders are pretrained and frozen, they expect full-length vectors, so we multiply the latent by a soft or binary mask, but do not zero out or truncate the tensor structure itself.
>
> * The masking operation $\odot$ retains the original dimensionality, but suppresses unselected dimensions via multiplication with mask values in $[0, 1]$ (or ${0,1}$ in the binary case after thresholding). This allows decoding to proceed without violating the input expectations of the pretrained VAEs.
>
> Then, as shown in Equation 8, the reconstruction terms in the total loss are:
>
> * $\ell_A(x_a, g_A(M_V \odot \tilde{z}_v))$, computed as MSE on log-Mel spectrograms,
>
> * $\ell_V(x_v, g_V(M_A \odot \tilde{z}_a))$, computed as frame-averaged pixel-wise MSE over video frames in the segment.
>
> These losses are computed between the original ground-truth modality and the output of the decoder fed with the masked latent from the other modality.
>
> To summarise:
>
> * The masked latent is used as input to the decoder during training.
>
> * Masking is applied via element-wise multiplication but retains full tensor shape, ensuring compatibility with convolutional decoders.
>
> * Reconstruction losses are computed using decoder outputs from masked latents.
>
> *To address the reviewer’s concern*, we will revise our presentation to make the following changes:
>
> * Revise Section 3.1 to explicitly state that masked latents are directly passed into decoders during training.
>
> * Clarify the meaning of “masked latent” in Equation 5 and in the surrounding text: i.e., masking preserves dimensionality via element-wise multiplication.
>
> * Clarify confusions around our earlier response (“decoding is not performed on physically masked latent”) to more accurately reflect the use of masked-but-full-shape latents.
>
> * Add explicit notes in the Implementation Details (Appendix C) describing the structure of the masked latents and how compatibility with convolutional VAEs is maintained.
>
> * Clarify the decoder loss terms in Eq. 8, including what modality is being reconstructed, what is being used to condition the decoder, and what loss function is applied.
>
> We hope this clears up the ambiguity, and we thank the reviewer again for the opportunity to clarify this key component of our method.

---

> ### Author Response · Authors · 2025-08-01
> **Thank you for the prompt response and providing further clarifications (Part 2/2)**
>
> **2. Providing more implementation details in the paper**
>
> We sincerely appreciate this valuable suggestion. The reviewer is absolutely right that including more practical details about the fine-tuning process would further improve the clarity of our work.
>
> While our theoretical analysis of masking was intended to motivate the selective alignment strategy, we recognize that readers may find the applied components, particularly the fine-tuning setup, more actionable for building upon our approach.
>
> Following your comments, in the revised manuscript we will:
>
> * Explicitly state in Section 3.1 that the video and audio diffusion models are fine-tuned using LoRA, and specify which layers are targeted.
>
> * Move some implementation-specific details from the Appendix to the main text, particularly those related to LoRA-based tuning and adapter interaction.
>
> * Clarify the distinction between Stage - alignment training and Stage - fine-tuning for generation so that the reader understands how each stage contributes to final output quality.
>
> * Optionally condense part of the theoretical justification in the main text with a summary and pointer to the Appendix, making room for more practical insights in the method section.
>
> We are grateful for this actionable feedback and believe these revisions will make our paper more accessible and useful to the community.
>
> ---
>
> **3. Change Two-stage into Three-stage for clarity**
>
> We thank the reviewer for this accurate and helpful observation. You are absolutely right. Our current terminology conflates the fine-tuning of diffusion models with inference, which can obscure the actual structure of our method. Following your comments, we will describe the process is consists of three distinct stages:
>
> * Masked Latent Adaptation (Stage I): Learning cross-modal alignment via adapter and masking modules.
>
> * Model Fine-tuning (Stage II): Fine-tuning the pretrained video and audio diffusion models using aligned latents to improve temporal and semantic consistency.
>
> * Multimodal Inference (Stage III): Generating video from text, projecting it via the trained adapter, and generating audio conditioned on both the text and aligned video latent.
>
> We appreciate the reviewer for highlighting this conceptual oversight, and we agree that revising this presentation will significantly improve clarity. Therefore, we will make the following revisions:
>
> * Revise Figure 3 and its caption to reflect a three-stage framework. We will clearly separate:
>
>     * Stage I: Masked Latent Adaptation
>
>     * Stage II: LoRA-based Fine-tuning
>
>     * Stage III: Cascaded Inference
>
> * Update Section 3 heading structure to introduce each of the three stages explicitly, renaming “Stage II: Cascaded Diffusion Inference” to “Stage II: Diffusion Model Fine-tuning” and introducing “Stage III: Inference” as a distinct subsection.
>
> * Add implementation details in Stage II describing the LoRA fine-tuning setup, including which model layers are adapted and why LoRA is chosen for parameter efficiency and modularity.
>
> * Update the Abstract and Introduction to briefly summarize the framework as a three-stage pipeline rather than a two-stage one.
>
> * Clarify evaluation timelines and training scope in the Experiments section, indicating which stages involve training and which are strictly for inference.
>
> These changes will ensure that the conceptual flow of our method is transparent and aligned with the actual implementation. We thank the reviewer again for this important clarification.
>
> ---
>
> We sincerely thank reviewer **wB3p** for your thoughtful comments and constructive engagement in the discussion. We hope that our responses and planned manuscript improvements have fully clarified the raised concerns and provided deeper insight into the design, implementation, and evaluation of our framework.
>
> Could you please let us know if you have any remaining questions, we are more than happy to continue the dialogue.

---

> ### Author Response · Authors · 2025-08-04
> **Thank you for your time and review.**
>
> Dear Reviewer wB3p,
>
> Thank you again for your valuable feedback.
>
> We hope you've had a chance to review our responses. If any questions or concerns remain, please let us know, and we'll promptly address them before the discussion period concludes.
>
> If our responses have satisfactorily resolved your concerns, we kindly ask if you would consider updating your score accordingly.
>
> Thank you for your time and consideration.
>
> The Authors

---

> > ### Comment · Reviewer_wB3p · 2025-08-04
> >
> > I would like to thanks the authors for the detailed answer and additional information. This indeed helps in understanding the proposed method.

---

### Official Review · Reviewer_iA4m · 2025-07-03

**Clarity:** 3
**Significance:** 3
**Originality:** 3
**Rating:** 5
**Confidence:** 4

**Summary:**

This paper addresses the challenge of generating temporally and semantically aligned audio-visual content from text (Text-to-Audio-Video, T2AV).
The authors propose SAVA, a novel framework for Selective Audio-Visual Alignment.
SAVA achieves state-of-the-art performance on standard T2AV benchmarks.

**Questions:**

1. The evaluation has limitations in baseline comparisons. In the T2AV experiment, the paper uses a cascaded text-to-video (T2V) and text-to-audio (T2A) approach but lacks a corresponding T2V+T2A baseline in Table 1 for direct comparison. Additionally, in the video-to-audio (V2A) experiment, the baselines are somewhat dated and limited in number. Including recent diffusion or flow-based models, as showcased in Figure 4 (e.g., Diff-Foley), would strengthen the evaluation’s comprehensiveness.

2. The ablation studies lack clarity on critical details. For example, in Table 3, it is unclear whether adapters trained under different alignment strategies required further fine-tuning of the downstream video and audio diffusion models. Providing explicit details on the training pipeline and fine-tuning process would enhance reproducibility and understanding of the results.

3. The paper does not include ablation studies exploring the impact of bypassing Stage 1’s Masked Latent Adaptation and directly training the adapter in Stage 2. Such an experiment could clarify whether the proposed masking strategy is essential or if a simpler adapter-based approach might yield comparable or better results, strengthening the justification for the two-stage design.


If the authors can provide clear and detailed responses to the concerns raised in **Weaknesses** and **Questions**, I would be willing to reconsider and potentially raise my overall score.

**Ethical Concerns:**

["NO or VERY MINOR ethics concerns only"]

**Final Justification:**

Recommended rating: "5 Accept"
Most of my concerns have been addressed.
In general, I think this article is acceptable.
In addition, I hope the author will open source the later code to facilitate the development of the community.

**Limitations:**

yes

**Quality:**

3

**Strengths And Weaknesses:**

**Strengths**

- Very good motivation. When audio and video are jointly generated, not all visual information and auditory information need to be interacted.

- The motivation is reasonable and the explanation is clear.



**Weaknesses**



1. Generate a one-way relationship between audio and video in the framework. The framework actually uses a cascaded generation, first generating video and then conditionally generating audio. The author did not give a good reason or experimental analysis for why this one-way relationship was adopted instead of the mutual influence between the two modalities.

2. The experimental design is unfair.

- some baselines are missing. The CoDi model is relatively old, and there are relatively new text-to-audio-video models, JavisDiT and SVG_baseline, which can be compared, but without comparison, it is difficult to judge the performance of the model.

- The existing baseline settings are not very reasonable. The video base model used by the author is CogVideoX, but the video model in Two-Streams and CasC-Diff uses AnimateDiff, so the comparison is not fair; the video-to-audio model does not use new ones such as DiffFoley MMAudio, and SpecVQGAN is relatively old.

3. Some suggestions on presentation. In formula 3, it is better to change the sign of z_v after it passes through q_V to distinguish it, otherwise it will cause ambiguity.

---

> ### Author Rebuttal · Authors · 2025-07-30
>
> We thank **Reviewer iA4m** for the thoughtful feedback and valuable suggestions, which have greatly helped us improve the clarity and depth of our work. Following are the responses to the questions and comments.
>
> ---
>
> **Q1: Current one-way design and potential bidirectional extension.**
>
> We thank the reviewer for raising this important point regarding the choice of a one-way relationship between video and audio in our framework.
>
> We wish to first clarify that the unidirectional generation process does not effect mutual influence between the two modalities, we employ a masked latent alignment mechanism during training. This mechanism encourages the model to learn a shared cross-modal latent space by optimizing for both audio-to-video and video-to-audio reconstruction. As a result, although generation proceeds in a single direction at inference time, the model is trained to encode the mutual semantic structure of both modalities. This training design ensures that relevant information from both audio and video is captured in the shared latent space, thereby preserving semantic alignment without requiring bidirectional generation during deployment.
>
> For practical consideration, visual content provides a more stable and structured prior for generating temporally aligned outputs. For example, the act of a drum being struck or a guitar being strummed is visually discrete and temporally grounded, providing strong cues for the expected acoustic outcome. Compared to audio, which may contain overlapping or ambiguous signals, visual representations often offer clearer event boundaries and object-specific dynamics. This structural clarity makes video a more reliable anchor modality when synthesizing aligned audio content. In addition, from a training perspective, finetuning the attention layers of a diffusion-based audio model to allow video-conditioning is substantially more efficient than modifying a large-scale video diffusion model, which further allows rapid adaptation to pretrained single-modal latent generative models.
>
> That said, we appreciate the reviewer’s suggestion and have conducted an extended experiment on VGGSound+ to explore the bidirectional generation setup. In this setting, we reverse the pipeline for video generation (audio generation pipeline is unchanged): raw matching audio is first generated using a text-conditioned AudioLDM [1] model, then encoded into a latent representation via the audio VAE. This latent is projected into the shared space using a lightweight audio-to-video adapter. We then fine-tuned the CogVideo [2] architecture by Lora-tuning its cross-attention layers to support dual conditioning from both text and the projected audio latent. This allows the video generator to incorporate semantic signals from both modalities during generation.
>
> |        | FVD  $\downarrow$     | FAV  $\downarrow$     | AVHScore $\uparrow$| CAVPSim $\uparrow$|
> |----------------|----------------|----------------|----------------|----------------|
> | Uni-directional  | 662.9 | 5.49  |0.206|0.165|
> | Bi-directional  | 701.4 | 5.49  |0.217|0.183|
>
> Our findings show that this bidirectional setup results in improved cross-modal alignment, as measured by both AVHScore and CAVPSim. However, we also observe a slight degradation in video quality, likely due to the noisier and less temporally structured nature of audio signals being used as primary conditioning input. Nonetheless, this result highlights a promising direction for future work, and we are actively exploring ways to refine this dual-conditioning scheme to achieve better trade-offs between alignment and generative fidelity.
>
> ---
>
> **Q2: Additional Baselines for T2AV and V2A Comparisons**
>
> Thank you for this important and detailed feedback.
>
> Regarding the use of CogVideoX [1] for our framework versus AnimateDiff [3] in some baselines: our primary goal was to demonstrate that our proposed alignment method improves audio-visual consistency regardless of the underlying video backbone. We adopted CogVideoX [1] to ensure high-fidelity video generation in our main method, and used AnimateDiff as baselines to remain consistent with  prior works such as TAVDiffusion [4]. However, we acknowledge that this discrepancy introduces a potential source of variance. To address this, we have conducted control experiments with corresponding T2V+T2A baselines to isolate the impact of framework design choices. Specifically, we construct the Two-Streams and CasC-Diff baselines using the same fine-tuned CogVideoX as the video backbone.
>
> |        | FVD  $\downarrow$     | FAV  $\downarrow$     | AVHScore $\uparrow$| CAVPSim $\uparrow$|
> |----------------|----------------|----------------|----------------|----------------|
> | Two-Streams  | - | 6.29  |0.061|0.092|
> | CasC-Diff | - | 7.12  |0.154|0.117|
> |Ours|662.9|5.49|0.206|0.165|
>
> The results demonstrate that the two-stream approach suffers from poor audio-visual alignment, while the vanilla cascade approach exhibits degradation in the quality of the generated audio conditioned on the generated video.
>
> Regarding the selected baselines, TAVDiffusion [4] and SeeingHearing [5] were both released in 2024 for T2AV and A2V generation. However, we recognize the importance of comparing against the most recent models. As suggested, we will include our concurrent work, JavisDiT [6], under a zero-shot setting on both the VGGSound+ and AudioCaps datasets. In addition, we compare the video-to-audio generative quality with Diff-Foley [7].
>
> |        | KL  $\downarrow$     | ISc  $\uparrow$     | FD $\downarrow$| FAD $\downarrow$|
> |----------------|----------------|----------------|----------------|----------------|
> | Diff-Foley  | 2.711 | 5.314  |37.262|6.861|
> | Ours  | 2.128 | 5.677  |39.534|6.155|
>
> In summary, to address your concerns regarding evaluation, we:
> * Add direct comparisons with JavisDiT in the T2AV setting, the results will be reported once ready.
> * Incorporate the recent V2A method, Diff-Foley.
> * Include the corresponding T2V+T2A results.
> * Clarify that our improvements primarily stem from enhanced cross-modal alignment rather than from superior unimodal generation models.
>
> ---
>
> **Q3: Ablation Study on Direct Alignment and Implementation Details**
>
> Table 3 in our manuscript reports the effect of bypassing Masked Latent Adaptation and directly training the adapter only. Our experimental results demonstrate that omitting the masking strategy significantly degrades audio generation quality and negatively impacts both the AVH score and CAVP similarity metrics. These findings underline the essential role of the masking strategy in effectively aligning the multimodal latents, thus justifying our proposed two-stage design. We will ensure this analysis and the corresponding insights are explicitly discussed in the revised manuscript.
>
> Regarding implementation details, our framework is built around a two-stage design. In the first stage, we train lightweight adapters to align latent representations between modalities. During this alignment stage, all components of the diffusion models, including the video and audio generators as well as their associated VAE encoders and decoders are kept frozen. The adapters are trained using the alignment objective introduced in Eq. 8 (Depending on the ablation, the input of mask varies. For direct alignment, the masks are set to all 1 to allow direct aligning, the sparsity losses become constant), to project one modality’s latent into the latent space of the other.
>
> Once the adapters are trained, we proceed to the second stage, where we enable conditional generation based on the aligned representations. To make the diffusion model responsive to the new conditioning signals that include both text and projected audio/video features, we apply LoRA-based fine-tuning only to the cross-attention layers of the target diffusion model. All other parameters remain frozen during this stage as well. This selective fine-tuning ensures efficient adaptation while preserving the pretrained capabilities of the base models.
>
> In the specific case of the ablation studies shown in Table 3, each row corresponds to a different alignment strategy applied to the adapter. For all these comparisons, the downstream diffusion models are held fixed after the initial LoRA-based tuning. That is, we evaluate each adapter variant in a zero-shot fashion without further training the diffusion model. This setup isolates the impact of the alignment strategy and allows for a clean comparison of adapter effectiveness.
>
> ---
>
> **Q4: Presentation of Formula 3**
>
> Thank you for pointing this out regarding Formula 3. We will adopt a distinct notation for this term in the revised manuscript to clearly distinguish its transformed representation.
>
> ---
>
> References:
>
> [1] Liu, H., et al. AudioLDM: Text-to-Audio Generation with Latent Diffusion Models. In International Conference on Machine Learning.
>
> [2] Yang, Z., et al. CogVideoX: Text-to-Video Diffusion Models with An Expert Transformer. In The Thirteenth International Conference on Learning Representations.
>
> [3] Guo, Y., et al. AnimateDiff: Animate Your Personalized Text-to-Image Diffusion Models without Specific Tuning. In The Twelfth International Conference on Learning Representations.
>
> [4] Mao, Y., et al. Tavgbench: Benchmarking text to audible-video generation. In Proceedings of the 32nd ACM International Conference on Multimedia.
>
> [5] Xing, Y., et al. "Seeing and hearing: Open-domain visual-audio generation with diffusion latent aligners." Proceedings of the IEEE/CVF Conference on Computer Vision and Pattern Recognition. 2024.
>
> [6] Liu, K., et al. "Javisdit: Joint audio-video diffusion transformer with hierarchical spatio-temporal prior synchronization." arXiv preprint arXiv:2503.23377 (2025).
>
> [7] Luo, S., et al. "Diff-foley: Synchronized video-to-audio synthesis with latent diffusion models." Advances in Neural Information Processing Systems 36 (2023).

---

> > ### Comment · Reviewer_iA4m · 2025-08-09
> > **Official Comment by Reviewer iA4m**
> >
> > I would like to thanks the authors for the detailed responds. I agree with Reviewer aSVy that the authors should provide more visualization examples. There are still some issues that need to be clarified:
> >
> >
> > - Could the authors please describe the scenarios in which failures primarily occur? Alternatively, could you provide a brief description of which scenarios might lead to hallucinations, producing sounds that differ significantly from real-world scenarios? What causes this?
> >
> > - Regarding the example of "man playing guitar" in Figure 5, does the generated sound strictly follow the string fluctuations like pitch and position in guitar, or is it an arbitrary guitar sound? How are these conditions controlled to ensure a reasonable tune?

---

> > > ### Author Response · Authors · 2025-08-09
> > > **Thank you for your response.**
> > >
> > > Dear Reviewer iA4m,
> > >
> > > Thank you for your insightful questions. We agree and commit to provide more visualizations in the updated manuscript.
> > >
> > > Below are our responses to your remaining questions:
> > >
> > > **1. Potential causes for failure**
> > >
> > > In our context, we assume “hallucination” includes both cases of generated audio that is not supported by the visual evidence or is implausible given the scene dynamics. Because our method aligns shared latent factors via masking, and then runs a video to audio cascade, failures typically arise when the shared signal is weak/ambiguous or contradictory. We suggest the following scenarios and the underlying causes below might potentially introduce hallucination:
> > >
> > > 1. Many simultaneous sources
> > >
> > >     Why: The mask must pick a minimal set of audio-relevant visual dimensions. When multiple sources compete, the selection can be unstable, leading to mixtures that emphasize the wrong source, or over-smoothed audio.
> > >
> > > 2. Domain shift
> > >
> > >     Why: When video latents are out-of-distribution, the learned mask may preserve spurious dimensions, the audio model then “rationalizes” the odd visuals with mismatched sounds.
> > >
> > > 3. Off-screen (or weakly visible) sound sources
> > >
> > >     Why: The shared visual–audio latent is intrinsically underdetermined (little/no visual evidence). The audio model falls back to priors learned from training data, which can sound plausible in general but mismatched for the specific scene.
> > >
> > > 4. Prompt–visual tension
> > >
> > >     Why: Our cascade prioritizes video at inference. When text and visuals diverge, audio follows the video, which can conflict with the textual intent and be perceived as “hallucination.”
> > >
> > > We will update the limitation discussion in our Appendix to case-analyse failure and hallucinations, where we categorize failures into the 4 buckets above with brief examples. Include qualitative cases (frame strips + spectrograms) and report the per-bucket metrics where feasible.
> > >
> > > ---
> > >
> > > **2. What is controlled and what is not**
> > >
> > > That's a great question. In the “man playing guitar” example, our system transfers instrument identity and rhythmic structure from the inputs, but it does not impose frame-accurate string or pitch control. Timbre is chiefly set by the text prompt (biasing the generator toward “guitar” acoustics), while the aligned video latent learned in Stage I provides the temporal scaffold: onset density, energy envelope, and strumming periodicity, so the spectrogram exhibits vertical striations that coincide with visible hand motions. We do not estimate fretting positions or track individual strings, nor do we apply any symbolic music constraints (key, chord progression, score). Consequently, the melodic content is plausible rather than strictly bound to the exact string fluctuations observed in the video.
> > >
> > > Reasonable musicality arises from the combination of these factors: the text prior anchors timbre, the video-conditioned latent enforces when sounds occur and their relative intensities, and LoRA fine-tuning of the audio model in Stage II improves responsiveness to the visual rhythm without dictating exact pitches. The result is a soundtrack that follows the visible strumming pattern and maintains a coherent “guitar” sound, while the precise pitch sequence remains unconstrained by explicit string-level supervision.
> > >
> > > ---
> > >
> > > We sincerely thank reviewer **iA4m** for your thoughtful comments and constructive engagement in the discussion. We hope that our responses and planned manuscript improvements have fully clarified the raised concerns and provided deeper insight into the design, implementation, and evaluation of our framework.
> > >
> > > Could you please let us know if you have any remaining questions, we are more than happy to continue the dialogue.

---

> ### Author Response · Authors · 2025-08-04
> **Thank you for your time and review.**
>
> Dear Reviewer iA4m,
>
> We would like to thank you again for your valuable feedback.
>
> We hope all your concerns have been fully resolved following our responses. If there are any additional issues or clarifications needed, please let us know, and we'll promptly address them before the discussion period concludes.
>
> If our responses have satisfactorily resolved your concerns, we kindly ask if you would consider updating your score accordingly.
>
> Thank you for your time and consideration.
>
> Best regards,
> The Authors

---

> ### Comment · Reviewer_iA4m · 2025-08-09
> **Official Comment by Reviewer iA4m**
>
> Thanks for the responses. Most of my concerns have been addressed.
>
>
> Overall, I think this article is acceptable.
> Consequently, I decide to raise my score.
> Note that the related issues and discusses are need to be clarified in the final version.
>
> Besides, I hope the author will public the source code of this work to facilitate the development of the T2AV community.

---

### Decision · Program_Chairs · 2025-09-17

**Decision:**

Accept (poster)

**Comment:**

This paper introduces a three-stage masked-latent-alignment framework for text-to-audio-video (T2AV) generation that selectively aligns audio and video modalities via cross-modal masking, achieving improved semantic and temporal coherence. The reviewers agree on the (1) novelty and theoretical grounding of the selective masking strategy, (2) effectiveness in cross-modal semantic alignment, (3) modular design that supports cascaded generation and integration with pretrained diffusion models, and (4) state-of-the-art performance on standard benchmarks. However, they note (1) inconsistencies and ambiguities in the method's presentation (especially regarding decoder behavior, loss computation, and stage descriptions), (2) insufficient qualitative analysis and visualizations of outputs and failure cases, and (3) a lack of clarity in the architectural and training details that required extensive rebuttal clarification. The authors’ follow-up responses addressed the major concerns: they clarified the masking and reconstruction mechanisms, outlined the exact decoder behavior during training, acknowledged the framework as a three-stage pipeline rather than two, and proposed revisions to improve exposition, while also conducting additional experiments including bidirectional generation and new baselines. The AC thus leans to accept this submission.